Corrected: Publisher correction

# CD146-HIF-1α hypoxic reprogramming drives vascular remodeling and pulmonary arterial hypertension

Yongting Luo [1], Xiao Teng[2], Lingling Zhang[3], Jianan Chen[4], Zheng Liu[4], Xuehui Chen[4], Shuai Zhao[4], Sai Yang[5], Jing Feng[4] & Xiyun Yan[4]

Pulmonary arterial hypertension (PAH) is a vascular remodeling disease of cardiopulmonary units. No cure is currently available due to an incomplete understanding of vascular remodeling. Here we identify CD146-hypoxia-inducible transcription factor 1 alpha (HIF-1α) cross-regulation as a key determinant in vascular remodeling and PAH pathogenesis. CD146 is markedly upregulated in pulmonary artery smooth muscle cells (PASMCs/SMCs) and in proportion to disease severity. CD146 expression and HIF-1α transcriptional program reinforce each other to physiologically enable PASMCs to adopt a more synthetic phenotype. Disruption of CD146-HIF-1α cross-talk by genetic ablation of *Cd146* in SMCs mitigates pulmonary vascular remodeling in chronic hypoxic mice. Strikingly, targeting of this axis with anti-CD146 antibodies alleviates established pulmonary hypertension (PH) and enhances cardiac function in two rodent models. This study provides mechanistic insights into hypoxic reprogramming that permits vascular remodeling, and thus provides proof of concept for anti-remodeling therapy for PAH through direct modulation of CD146-HIF-1α cross-regulation.

[1] Beijing Advanced Innovation Center for Food Nutrition and Human Health, China Agricultural University, Yuanmingyuan West Road 2, 100193 Beijing, China. [2] State Key Laboratory of Cardiovascular Diseases, Fuwai Hospital, National Center for Cardiovascular Diseases, Chinese Academy of Medical Sciences and Peking Union Medical College, No. 167 North Lishi Road, 100037 Beijing, China. [3] Department of Rheumatology and Clinical Immunology, Peking Union Medical College Hospital, Chinese Academy of Medical Sciences & Peking Union Medical College, No. 1 Shuaifuyuan, 100730 Beijing, China. [4] Key Laboratory of Protein and Peptide Pharmaceutical, Institute of Biophysics, Chinese Academy of Sciences, 15 Datun Road, 100101 Beijing, China. [5] Laboratory Animal Research Center, Institute of Biophysics, Chinese Academy of Sciences, 15 Datun Road, 100101 Beijing, China. Correspondence and requests for materials should be addressed to Y.L. (email: luo_yongting@163.com) or to X.Y. (email: yanxy@ibp.ac.cn)

Pulmonary arterial hypertension (PAH) is a group of progressive diseases of the cardiopulmonary unit characterized by irreversible right ventricular failure, ultimately resulting in premature death. A common pathological hallmark of PAH is vascular remodeling, manifested by medial thickening, enhanced muscularity of pulmonary arteries (PAs), and uncontrolled expansion of PASMCs[1,2]. This structural alterations of small PAs contributes to the progressively increased artery resistance and right ventricular dilatation[3]. Despite substantial advances in PAH treatment in the last decade, PAH diagnosis is commonly associated with extremely poor prognosis, worse than many cancers[4]. At present, few effective therapeutic options other than lung transplantation are available. Existing treatments predominantly target vasoconstriction through vasodilators involving prostanoids, endothelin receptor blockers, and/or phosphodiesterase-5 inhibitors[5,6]. Although these medications provide symptomatic relief, they only modestly improve survival as they are unable to prevent excessive SMC burden and occlusive remodeling[7]. This limited amelioration effect of existing treatments underscores the urgent need for identifying suitable targets driving the maladaptive remodeling as a first step to develop effective anti-PAH drugs.

The pulmonary hypoxic response is an evolutionarily ancient and conserved stress response triggered by reduced alveolar oxygen availability. Chronic hypoxia has emerged as a well-established independent cause of vascular remodeling that also exacerbates pulmonary hypertension (PH) owing to other etiologies[8]. Among all hypoxia-responsive factors, HIF-1α functions as a master regulator of oxygen homeostasis and hypoxic adaptation in the lung. During PH pathogenesis, aberrant HIF-1α activation determines the expansion of PASMCs and pulmonary vascular remodeling[9,10]. For instance, mice with *Hif1a* heterozygous inactivation or inducible deletion in SMCs exhibit reduced vascular remodeling and blunted PASMC hypertrophy in chronic hypoxic mice[11,12]. In support of this observation, HIF-1α has been implicated in clonal expansion of hypoxia-primed SMC progenitors that govern muscularization during hypoxic PH[13]. In addition, HIF-1α also functions as an intrinsic pathogenic determinant in chemically induced and genetic forms of PH[14–16]. These studies indicate that controlling HIF-1α-driven vascular remodeling may provide a promising avenue for PAH pharmacotherapies. In spite of these findings, defining the mechanism governing HIF-1α activation in PASMC, thereby developing effective therapeutic options for PAH remains largely unexplored[17].

CD146 is constitutively and stably expressed in SMCs constructing various arteries from multiple organs through embryonic development[18]. During development as well as during vascular repair in adults, CD146 plays a role in cell differentiation, proliferation, as well as turnover of aortic SMC[19]. Nevertheless, the precise involvement and contribution of CD146 in the pathogenesis of vascular diseases pertaining arterial remodeling, such as PAH, remain to be established. Notably, the apparent functional role of CD146 in vascular diseases, such as atherosclerosis[20], multiple sclerosis[21], systemic sclerosis[22], and cancer[23], is dependent on its capacity in promoting cell proliferation, inflammation and fibrosis, the prominent hallmarks of PAH. Moreover, studies in vascular cells suggested that a possible link exists between CD146 expression and HIF-1α activation[24–26]. Given such close association of CD146 with PAH features and the broad impact of CD146 in vascular disorders, we aimed to determine the connection between CD146 and HIF-1α in vascular remodeling, and to further explore its potential as an anti-remodeling target for PAH therapy.

Here we demonstrate that a cross-regulation between CD146 and HIF-1α in PASMCs triggers pulmonary vascular remodeling. Disruption of CD146-HIF-1α axis in PASMC blunts vascular remodeling and produces a marked attenuation of PH. These findings uncover previously uncharacterized hypoxic reprogramming during vascular remodeling, and suggest that blocking of CD146-HIF-1α cross-regulation represents a suitable avenue for developing effective therapies against PAH.

## Results

**CD146 elevation in PASMCs correlates with PH severity**. To investigate whether CD146 is involved in the development of PH, we tested its expression pattern in normal and pulmonary hypertensive lungs by examining lung samples from mice with hypoxia-induced PH and rats with monocrotaline (MCT)-induced PH compared with their respective controls. We observed elevated CD146 mRNA (Fig. 1a) and protein (Fig. 1b) levels in both mouse and rat PH lungs compared with control lung tissues. To evaluate whether CD146 expression is associated with disease severity, we studied rodent lung tissues during the development of both hypoxia- and MCT-induced PH. We found that CD146 mRNA (Fig. 1c) and protein (Fig. 1d) levels progressively increased at serial time points after PH induction. Moreover, Pearson correlation analysis showed that lung *Cd146* mRNA levels were positively correlated with the levels of right ventricular systolic pressure (RVSP), a commonly used key indicator of worsening disease severity (Fig. 1e, f).

To study whether the upregulation of CD146 is confined to PAs, we first compared CD146 expression in PAs and non-PAs of lung tissue from mice exposed to normoxia or hypoxia. CD146 was mainly observed in PA fraction but not in non-PA fraction, and hypoxia elevated CD146 expression in PAs (Fig. 1g). We then examined its expression pattern in PAs using 3D-reconstruction post-confocal imaging of lung sections from healthy subjects. CD146 expression in PAs with differential vessel diameters (30–100 μm) was mainly observed in PASMCs as identified by co-staining with α-smooth muscle actin (αSMA) and SMHHC, but undetectable in pulmonary artery endothelial cells (PAECs/ECs) with complete SMC ensheathment (Supplementary Fig. 1). In addition, hypoxia-induced CD146 elevation was mainly occurred in PASMCs, whereas CD146 expression in PAECs was not detectable (Fig. 1h). To confirm whether CD146 is specifically upregulated in PASMCs of PH lungs, we performed immunofluorescent staining of lung tissues. We observed that, in contrast to healthy lungs, CD146 expression was upregulated and prominently confined to the medial layer of PASMCs in the remodeled small pulmonary arteries as identified by co-staining with αSMA (Fig. 1i, j). However, CD146 expression in the neointima was almost undetectable (Fig. 1k). The analysis of the colocalization of CD146 signal with αSMA or CD31 signal further confirmed the specific expression of CD146 within medial layer of PH lungs (Fig. 1l), indicating that the upregulation of CD146 in PASMCs is a common feature of PH. To test whether CD146 elevation in PASMCs is proportional to PH disease severity, we isolated PASMCs from both mice and rats during PH progression. It was found that CD146 protein levels gradually increased during hypoxia- or MCT induction (Fig. 1m, n). Together, these results indicated that elevated CD146 in PASMCs represents a reliable marker for PH severity.

**HIF-1α induces CD146 expression in PASMCs**. As hypoxia elevates CD146 mRNA, and tissue hypoxia is a direct driver of pathobiology in various presentations of PH[14–16], we next explored the potential role of hypoxia in regulating CD146 expression. We first confirmed the effect of hypoxia on CD146 transcripts in human PASMCs (hPASMCs) exposed to differential contents of $O_2$ (1%, 5%, or 21%) for indicated times.

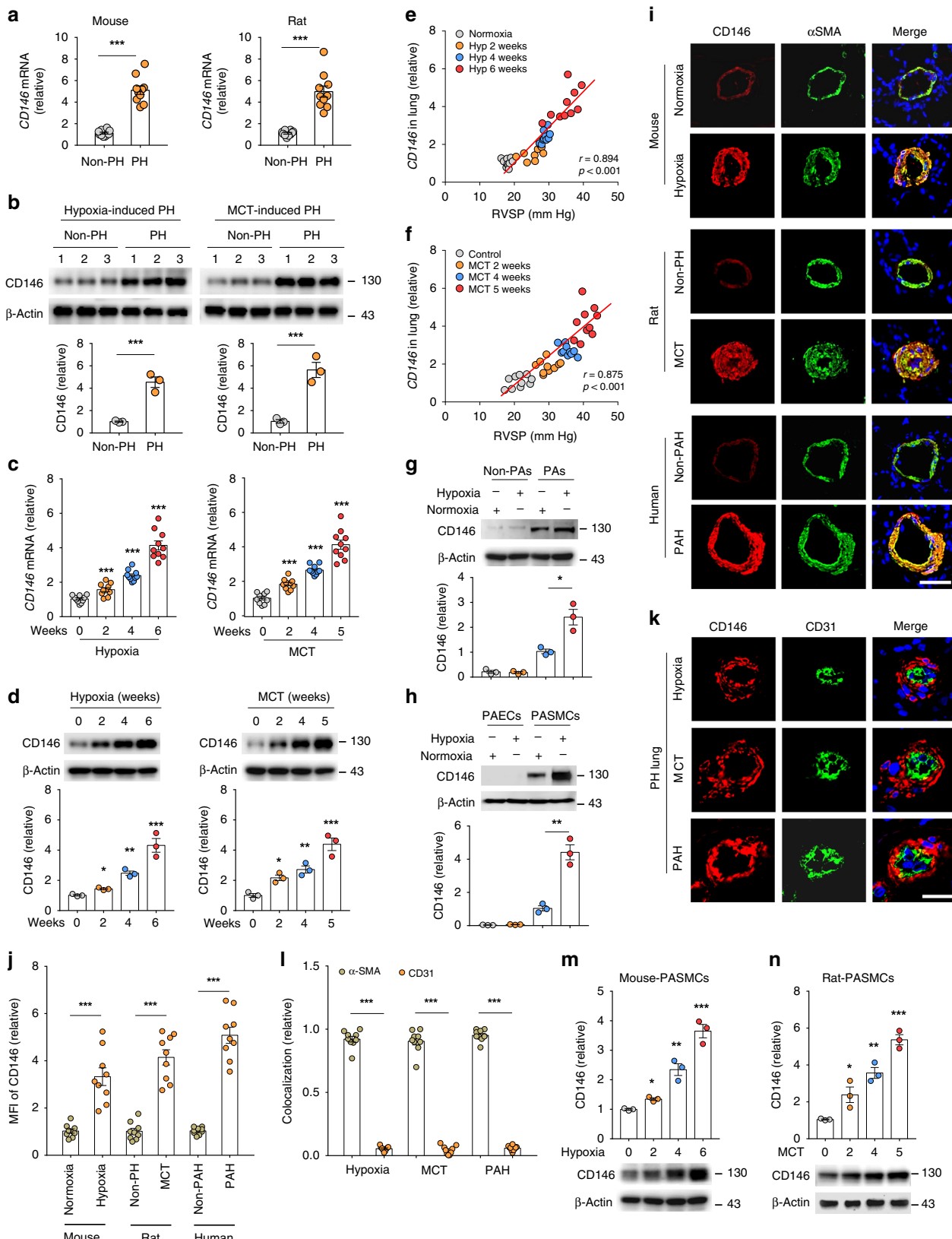

Increased CD146 expression at both mRNA (Fig. 2a, b) and protein (Fig. 2c) levels were observed in hypoxic cells in a $O_2$ dose- and time-dependent manner. To determine how CD146 is induced by hypoxia, we knocked down three well-known hypoxia-responsible transcriptional factors, HIF-1α, HIF-2α, and TP53, in 293 T cells. Knockdown of *HIF1A*, but not *HIF2A* or

*TP53*, markedly reduced hypoxia-induced CD146 upregulation (Fig. 2d, e). Conversely, treatment of hPASMCs with hypoxia, $CoCl_2$, DMOG or DFX, inducers of HIF-1α expression[27], results in CD146 upregulation (Fig. 2f, g). Consistent with hypoxia induction, ectopic expression of HIF-1α increased CD146 level in a dose-dependent manner (Fig. 2h, i), confirming that HIF-1α is

**Fig. 1** CD146 elevation in PASMCs correlates with PH severity. **a**, **b** CD146 mRNA **a** or protein **b** expression in lungs with hypoxia-induced PH in mice and with MCT-induced PH in rats. Below in **b**, relative CD146 expression. **c**, **d** CD146 mRNA **c** or protein **d** expression in the lungs of mice during development of hypoxia-induced PH or of rats during development of MCT-induced PH. Below in **d**, relative CD146 expression. $n = 3$ **b**, **d** or $n = 10$ **a**, **c** lungs for each group. **e**, **f** Pearson comparison analyses showing the correlation between *Cd146* mRNA levels (normalized to β-actin) in lung tissues and RVSP of mice during development of hypoxia-induced PH **e** or of rats during development of MCT-induced PH **f** ($n = 10$ animals for each time point). **g**, **h** Top, mice were exposed to hypoxia for 4 weeks. The expression of CD146 in PAs and non-PAs fractions **g** or in isolated PAECs and PASMCs **h** was detected by western blotting (WB). Bottom, quantification of CD146 expression ($n = 3$ independent experiments). **i–k** Representative immunofluorescence of CD146 (red) and αSMA (green) **i** or CD31 (green) **k** in small pulmonary arteries from mouse, rat and human lungs without or with PH. Nuclei are counterstained with DAPI (blue). Scale bars, 50 μm. The images are representative of nine arteries per group. **j** Quantification of the mean fluorescent intensity (MFI) of CD146 ($n = 9$ arteries per group). **l** Quantification of the colocalization of CD146 and αSMA signal or CD146 and CD31 signal in **i** and **k**. **m**, **n** Bottom, CD146 protein expression in PASMCs isolated from mice during development of hypoxia-induced PH **m** or from rats during development of MCT-induced PH **n**. Top, relative CD146 expression ($n = 3$ independent experiments). In all statistical plots, the results are expressed as mean ± s.e.m. $P < 0.05$, **$P < 0.01$, ***$P < 0.001$. One-way ANOVA with Bonferroni post hoc analysis **a–c** or two-tailed Student's $t$ test **d**, **g**, **h**, **j**, **l**, **m**, **n**. All WB represent data from three **b**, **d**, **g**, **h**, **m**, **n** independent experiments. Source data are provided as a Source Data file

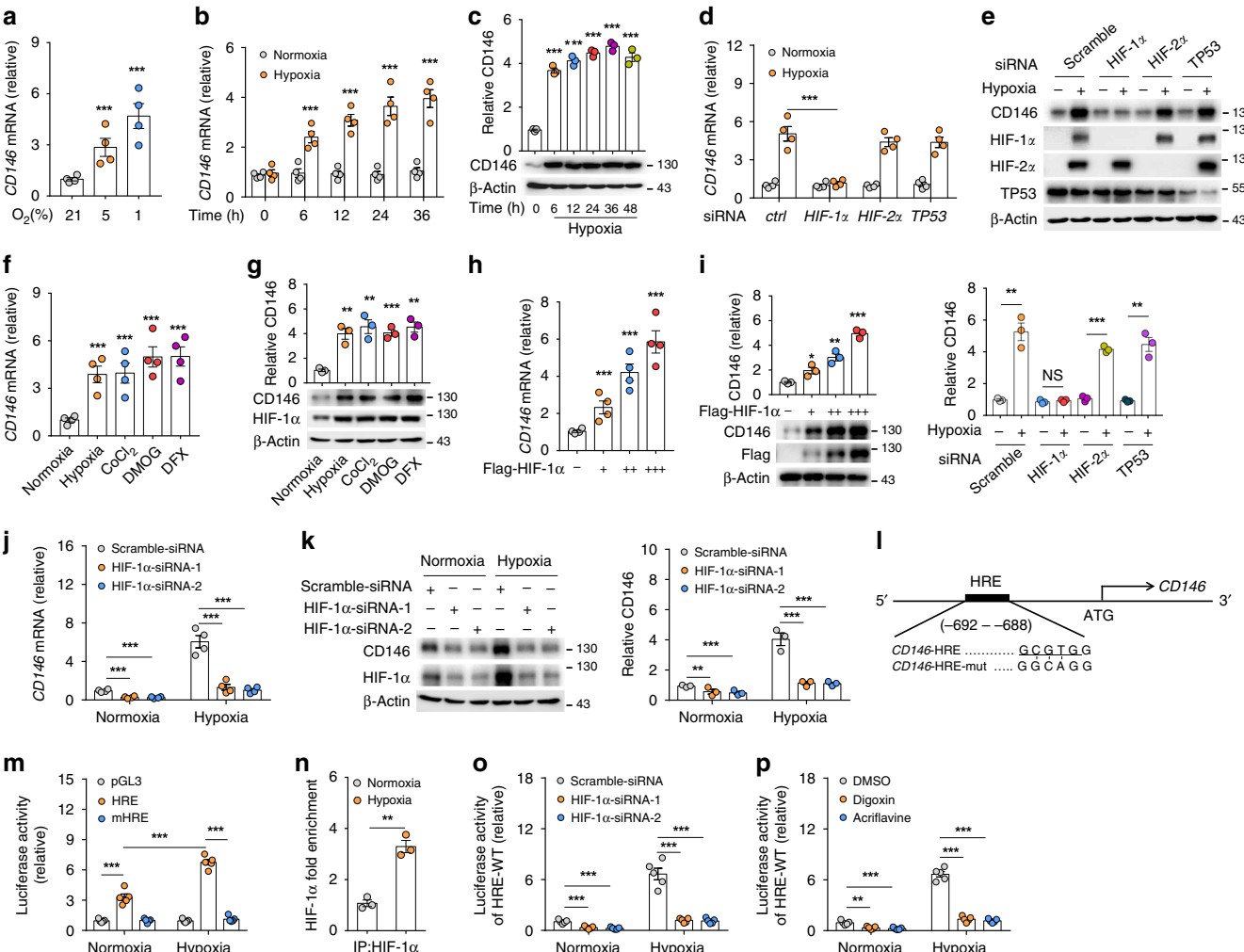

**Fig. 2** HIF-1α induces CD146 expression in PASMCs. **a**, **b** Relative mRNA levels of *CD146* in hPASMCs exposed to normoxia or hypoxia for indicated times. **c** CD146 protein expression in hPASMCs exposed to 21% or 1% O₂ for indicated times. **d**, **e** 293 T cells were transfected with siRNA targeting *HIF1A*, *HIF2A*, or *TP53* for 24 h before exposed to 1% or 21% O₂ for another 24 h. CD146 expression at mRNA **d** or protein **e** level was measured. **f**, **g** Human PASMCs were treated with hypoxia, CoCl₂ (100 μM), DMOG (100 μM) or DFX (50 μM) for 24 h before CD146 mRNA **f** or protein **g** level was measured. **h–k** Human PASMCs were transfected with HIF-1α-expressing plasmid **h**, **i** or HIF-1α-siRNA **j**, **k** for 24 h before CD146 expression was measured. **l** Design of luciferase reporter vector containing human *CD146* promoter that includes an HRE or a mutant HRE (mHRE). **m** Dual luciferase assay of putative HIF-1α-binding sites in *CD146* promoter. The luciferase activity in pGL3 transfected construct was set as 1. **n** ChIP using ChIP-grade HIF-1α antibody and quantification of the enrichment of HIF-1α binding to *CD146* promoter after hypoxia stimulation ($n = 3$ per group). **o**, **p** Luciferase assay of HRE-WT reporter in human PASMCs transfected with HIF-1α siRNAs **o**, or treated with HIF-1α inhibitor digoxin or acriflavine. **p** Reporter activity was measured and plotted after normalizing with respect to Renilla luciferase activity. $n = 3$ **n**, $n = 4$ **a**, **b**, **d**, **f**, **h**, **j**, or $n = 5$ **m**, **o**, **p** biological replicates for each group. All WB represent data from three **c**, **e**, **g**, **i**, **k** independent experiments. In all statistical plots, the results are expressed as mean ± s.e.m. *$P < 0.05$, *$P < 0.01$, ***$P < 0.001$. Two-tailed Student's $t$ test. Source data are provided as a Source Data file

essential for inducing CD146 transcription under hypoxia. Furthermore, knockdown of HIF-1α using siRNA in hPASMCs inhibited CD146 expression under both normoxia and hypoxia conditions (Fig. 2j, k). This result was consistent with previous studies showing that under normoxia, cultured PASMCs also express low levels of HIF-1α[28–30].

The transcriptional activator HIF-1α acts by binding to hypoxia response element (HRE) of target genes upon hypoxia to initiate extensive transcriptional cascades[31]. Here, we utilized a HRE luciferase reporter system to examine how HIF-1α affects CD146 expression. Further examination of *CD146* gene revealed a HRE encoding HIF-1α-binding motif (Fig. 2l) located 692 bp upstream of *CD146* transcription start site. We cloned a 1 kb fragment of the 5′ region of *CD146* containing this HRE into the luciferase reporter vector pGL3. hPASMCs transfected with the *CD146* HRE reporter vector demonstrated significantly increased luciferase activity after exposure to hypoxia, whereas the luciferase activity of cells transfected with the mutant HRE vector remained at basal normoxia levels (Fig. 2m). Chromatin immunoprecipitation (ChIP) assays showed the increased binding of HIF-1α to this *CD146* HRE after hypoxia exposure (Fig. 2n). Moreover, blocking HIF-1α expression by siRNA or inhibiting HIF-1α activity by digoxin or acriflavine (potent HIF-1α inhibitors) abolished hypoxia-induced HRE luciferase activity (Fig. 2o, p), further confirming the enrichment of HIF-1α bound to *CD146* HRE under hypoxia.

**CD146 promotes HIF-1α transcriptional program through NF-κB**. Given that CD146 elevation correlates with PH progression, and CD146 possibly links to hypoxic signaling, we hypothesized that CD146 initiates HIF-1α transcriptional program in PASMCs under hypoxia. To prove this, we first isolated primary PASMCs from the lung of *Cd146*[WT] and *Cd146*[SMC-KO] mice (Supplementary Fig. 2a–c). These cells expressed smooth muscle cell markers, but were negative for EC marker, demonstrating the high purity of PASMC culture (Supplementary Fig. 2d). Further molecular analysis showed that both steady-state and hypoxia-induced HIF-1α expression at both mRNA and protein levels were reduced in *Cd146*-deficient PASMCs (Fig. 3a, b). As shown in Fig. 3c, in the presence of actinomycin D, *Hif1a* transcript levels were only slightly decreased, suggesting that HIF-1α transcription rather than RNA stability accounts for the decrease in *Hif1a* mRNA. In agreement with this observation, we found that HIF-1α promoter activity (Fig. 3d) and HIF-1α DNA binding to its target genes, including *carbonic anhydrase IX* (*Ca9*) and *plasminogen activator inhibitor-1* (*Pai1*), were also reduced (Fig. 3e, f). In addition, the decreased HRE reporter activity in *CD146*-deficient cells was rescued by ectopic expression of HIF-1α (Fig. 3g). Moreover, when HIF-1α activity was inhibited by treatment with digoxin, *Cd146* knockout failed to affect wild-type (WT) reporter activity under hypoxia (Fig. 3h). Notably, hypoxia-induced HIF-1α expression, HRE reporter activity and transcriptional activity were inhibited in *CD146*-silenced but not control hPASMCs (Supplementary Fig. 3), further support the idea that CD146 contributes to regulate HIF-1α transcriptional program.

On the basis of the observation that CD146 regulates *HIF1A* mRNA, we next explored the mechanism of how CD146 triggers *HIF1A* transcription in PASMCs. By re-examining how transcriptional factors downstream of CD146 are regulated in response to hypoxic conditions, we found that hypoxia is able to activate NF-κB, which requires CD146 expression in both human and mouse PASMCs (Supplementary Fig. 4). We next examined whether NF-κB is involved in *HIF1A* transcription. Analysis of *HIF1A* promoter for transcriptional factor binding sites revealed a putative NF-κB-binding motif (Supplementary

Fig. 5a), which has been reported to regulate *HIF1A* transcription in HEK-293 cells[32]. Luciferase reporter assay showed that *HIF1A* promoter activity was robustly enhanced under hypoxic culturing conditions, and mutations of this binding site abolished luciferase activity (Fig. 3i). Importantly, the increased *HIF1A* promoter activity (Supplementary Fig. 5b) as well as hypoxia-induced binding of NF-κB to *HIF1A* promoter (Fig. 3j) was abolished by adding NF-κB inhibitor BAY 11–7082 into the culture medium. These results suggested that NF-κB is one of the transcription factors essential for HIF-1α protein accumulation in PASMCs.

To determine whether CD146-dependent NF-κB activation is involved in the HIF-1α-dependent transcriptional program, *Cd146*-null PASMCs were transfected with CD146-expressing plasmid, and then exposed to hypoxia. We found that rescuing CD146 expression induced NF-κB activation and HIF-1α expression, while blocking NF-κB activation by either knockdown *p65* (Fig. 3k–m and Supplementary Fig. 6a) or treatment with NF-κB inhibitor diminished CD146-induced HIF-1α expression and transcriptional activity (Fig. 3n–p and Supplementary Fig. 6b), indicating that CD146-induced NF-κB activation is critical for HIF-1α-induced transcriptional program in PASMCs. As NF-κB activation requires CD146 dimerization[33,34], we next investigated the impact of CD146 dimerization on HIF-1α activity. The results showed that CD146-induced NF-κB activation, as well as HIF-1α expression and activity were inhibited by ectopic expression of dimer defective mutant CD146-C452A or CD146-C499A (Fig. 3q–s and Supplementary Fig. 6c), suggesting that CD146 dimerization is critical for NF-κB-mediated HIF-1α activation. To confirm the role of CD146 dimerization on hypoxic signaling, we further applied anti-CD146 monoclonal antibody AA98 that abrogates CD146 dimerization[33]. As shown in Fig. 3t–v and Supplementary Fig. 6d, hypoxia-induced NF-κB activation and HIF-1α activity were impaired upon AA98 treatment, indicating the requirement of CD146 dimerization in HIF-1α transcriptional program in PASMCs.

Together, these results indicate that HIF-1α upregulates CD146 through direct binding to its promoter region. The accumulation of CD146, in turn, promotes HIF-1α transcriptional program through activation of NF-κB. Therefore, the cross-regulation of CD146-HIF-1α forms a positive feedback loop crucial for hypoxic reprogramming of PASMCs.

**CD146-HIF-1α axis promotes a synthetic phenotype in PASMCs**. HIF pathway drives the pathogenesis of PH by triggering vascular remodeling during disease progression. In this regard, the hyper-proliferation, migration, and a switch from contractile to synthetic phenotype of PASMCs in the media layer is the key event in vascular remodeling[35]. We then examined the role of CD146-HIF-1α cross-regulation in the phenotypic switch of PASMCs driven by hypoxia. We first investigated whether CD146-HIF-1α regulates proliferation and apoptosis of PASMCs under hypoxia. We found that both knockdown of *CD146* (Fig. 4a, b and Supplementary Fig. 7a) as well as *Cd146* deficiency (Fig. 4c, d and Supplementary Fig. 7b) inhibited hypoxia-induced cell proliferation, whereas simultaneously promoting cell apoptosis (as indicated by apoptosis-related proteins, including Bcl2, Bcl-xL, cleaved caspase 3 and cleaved caspase 9). Rescuing either CD146 or HIF-1α expression in *CD146*-silenced cells restored these effects. In agreement with these results, blocking HIF-1α activation by anti-CD146 AA98 suppressed hypoxia-induced cell proliferation, while promoted apoptosis, which was restored by HIF-1α ectopic expression in both human and mouse PASMCs (Fig. 4e, f and Supplementary Fig. 7c, 8a & b), indicating that CD146-HIF-1α cross-regulation has a major role in hypoxia-induced proliferation while inhibiting apoptosis in PASMCs.

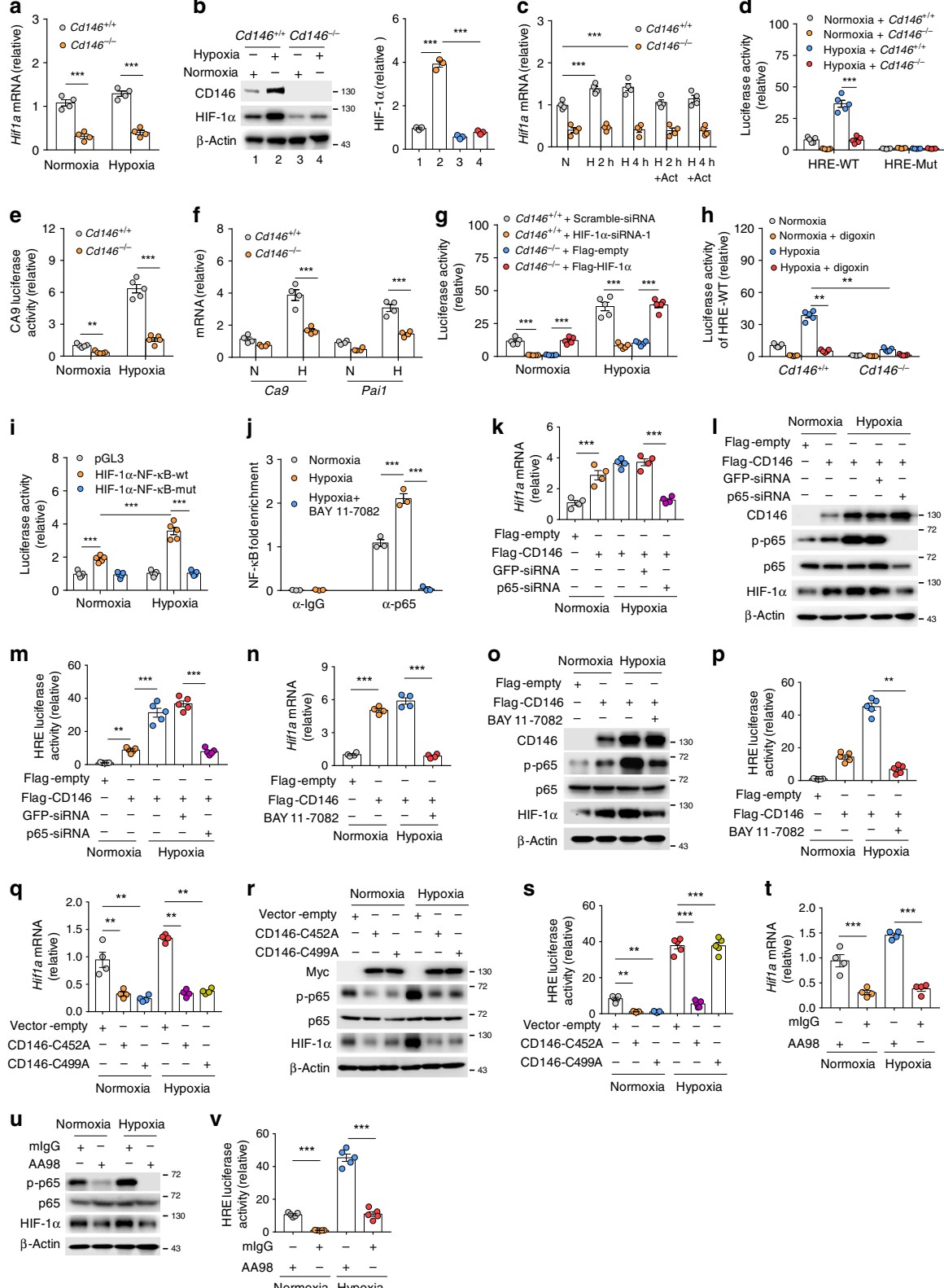

Muscularization of previously unmuscularized pulmonary arterioles requires the migration of SMCs to the distal part of pulmonary vessels[36]. We next determined whether CD146-HIF-1α axis is required for hypoxia-induced PASMC migration. We found that the reduced migratory capability of PASMCs after CD146 downregulation (Fig. 4g, h) or AA98 treatment (Fig. 4i and Supplementary Fig. 8c) was restored by rescuing the

expression of CD146 or HIF-1α, indicating that hypoxia-induced cell migration relied heavily on CD146-HIF-1α axis. In addition, the increased expression of contractile markers (including *SM22A*, *SMMHC*, *SMTN*, and *CNN1*) or decreased expression of synthetic PASMC markers (including *COL1A1*, *FN1*, and *VIM*) after downregulating or blocking CD146 was restored by rescuing the expression of CD146 or HIF-1α in

**Fig. 3** CD146 promotes HIF-1α transcriptional program through NF-κB. **a**, **b** HIF-1α mRNA **a** or protein **b** expression in murine PASMCs ($Cd146^{+/+}$ and $Cd146^{-/-}$) under normoxia or hypoxia. **c** Quantification of HIF-1α transcript in actinomycin D-treated PASMCs. **d–f** Luciferase assay of HRE-WT/HRE-Mut reporter **d** or Ca9 reporter **e**, or mRNA levels of Ca9 and Pai1 **f** in PASMCs ($Cd146^{+/+}$ and $Cd146^{-/-}$) under normoxia or hypoxia. **g**, **h** Luciferase assay of HRE-WT in $Cd146^{+/+}$ PASMCs transfected with HIF-1α-siRNA or in $Cdd146^{-/-}$ PASMCs transfected with HIF-1α-expressing plasmid **g**, or treated with goxin **h**. **i** Dual luciferase assay of HIF1A promoter with the putative NF-κB-binding site or mutant site. **j** Quantification of ChIP assay of NF-κB binding to HIF1A promoter in human PASAMCs cultured under normoxic or hypoxic conditions in the presence or absence of NF-κB inhibitor BAY 11–7082. **k–m** HIF-1α mRNA **k** or protein **l** expression or reporter activity of HRE **m** in $Cd146^{-/-}$ PASMCs transfected with CD146-expressing plasmid or p65-siRNA. **n–p** HIF-1α mRNA **n** or protein **o** expression or reporter activity of HRE **p** in $Cd146^{+/+}$ PASMCs treated with or without NF-κB inhibitor BAY 11–7082. **q–s** HIF-1α mRNA **q** or protein **r** expression or reporter activity of HRE **s** in human PASMCs transfected with CD146 dimer mutant-expressing plasmids. **t–v** HIF-1α mRNA **n** or protein **o** expression or reporter activity of HRE **p** in human PASMCs treated with anti-CD146 AA98 or mIgG (50 µg/ml). $n = 3$ **j**, $n = 4$ **a**, **c**, **f**, **k**, **n**, **q**, **t**, or $n = 5$ **d**, **e**, **g**, **h**, **i**, **m**, **p**, **s**, **v** biological replicates for each group. All WB represent data from three **b**, **i**, **o**, **r**, **u** independent experiments. In all statistical plots, the results are expressed as mean ± s.e.m. **P < 0.01, ***P < 0.001. Two-tailed Student's t test. Source data are provided as a Source Data file

response to hypoxia (Fig. 4j–o and Supplementary Fig. 8d & e). Together, these results suggest that under hypoxic conditions, CD146-HIF-1α axis enables these cells to proliferate, migrate, lose expression of contractile markers, and enhance expression of synthetic markers in PASMCs.

**Disruption of CD146-HIF-1α in SMCs attenuates hypoxic PH.** To provide direct genetic evidence that disrupting CD146-HIF-1α axis in SMCs attenuates pulmonary vascular remodeling, we studied the development of hypoxic PH in SMC $Cd146$ knockout mice (Fig. 5a). The upregulation of CD146, p-p65, and HIF-1α (Fig. 5b), as well as HIF-1α target genes (Fig. 5c) observed in WT lungs during hypoxia was inhibited in PASMCs from $Cd146^{SMC-KO}$ mice, confirming that CD146 is essential for proper functioning of the HIF-1α transcriptional program in vivo.

After exposure to hypoxic conditions, WT mice developed significant elevation in RVSP (Fig. 5d), total pulmonary vascular resistance index (TPVRI) (Fig. 5e) and RV hypertrophy (ratio of the weight of free right ventricular wall to the weight of left ventricular wall plus septum, RV/LV + S) (Fig. 5f) compared with that in normoxic controls. Intriguingly, $Cd146^{SMC-KO}$ mice displayed a much reduced PH phenotype compared with WT mice, as determined by RVSP, TPVRI, and RV/LV + S under hypoxia (Fig. 5d–f). We observed no changes in either SAP (systemic arterial pressure) or body weight between the two groups (Fig. 5g, h). In addition, $Cd146^{SMC-KO}$ mice also showed significantly extended PA acceleration time (PAAT), shortened RV internal diameter (RVID) and RV free wall thickness (RVWT), higher tricuspid annular plane systolic excursion (TAPSE), and increased cardiac output (CO) and cardiac index (CI) as determined by echocardiography (Fig. 5i, j). However, under normoxia, we failed to detect any changes in these parameters (PAAT, RVID, RVWT, TAPSE, CO, and CI) between the two genotypes.

Similarly to the observed hemodynamic changes, $Cd146^{SMC-KO}$ mice maintained in hypoxia exhibited reduced pulmonary vascular wall thickness (Fig. 5k, l), luminal stenosis (Fig. 5m), and muscularization (Fig. 5n), suggesting attenuated pulmonary vascular remodeling in response to chronic hypoxia. In addition, we observed fewer proliferating PASMCs (Fig. 5o) and more apoptotic cells (Fig. 5p) in the remodeled small PAs of $Cd146^{SMC-KO}$ mice compared with WT controls. These results indicate that the deactivation of CD146-HIF-1α axis in SMC on hypoxic PH results in both antiproliferative and proapoptotic effects on PASMCs. Together, these data indicated that the CD146-HIF-1α axis critically modulates pulmonary vascular remodeling in PASMCs.

To examine whether the attenuated development of PH in $Cd146^{SMC-KO}$ mice can be attributed to the alterations in pulmonary vessel myogenic tone and hemodynamics, changes in vascular integrity/permeability or inhibition of pulmonary vasoconstriction, we performed four sets of experiments. First, we compared pulmonary vessel density and vascular wall thickness in $Cd146^{SMC-KO}$ mice, with WT littermates as control. We found no differences in the ratio of the number of vessels vs. alveoli (Supplementary Fig. 9a) and pulmonary vascular wall thickness (Fig. 5k, l & n) between the two groups. Second, no changes were detected in the TPVRI (Fig. 5e), PAAT and pulmonary CO/CI (Fig. 5i, j), suggesting that the pulmonary hemodynamics were comparable in the two groups. Third, we found no differences in the Wet/Dry weight ratio and water content of the lung, suggesting that the integrity of pulmonary vessels remained unaffected by knockout of $Cd146$ in SMC (Supplementary Fig. 9b & c). Moreover, to further assess the vascular permeability, we performed diffusion assays using different tracers. Although our previous studies showed that $Cd146$ deletion in perivascular cells results in impaired vascular integrity in the brain[18], the permeability of vessels in the lung to Evan's blue dye (-labelled albumin) and different sizes of fluorescent dextrans (70 kD and 4 kD) was similar between the two genotypes (Supplementary Fig. 9d–f), suggesting that CD146 has different roles in vascular integrity in brain and pulmonary circulations. Fourth, we compared agonist- or hypoxia-induced vasoconstriction of isolated intraacinar small arteries from the two genotypes by morphometric analysis. By measuring the relative luminal area, we observed that neither hypoxia- or U46619-induced vasoconstriction, as determined by measuring the relative luminal area, was changed between the two groups (Supplementary Fig. 10a & b). Together, these results suggest that CD146 expressed on SMC is mainly involved in pulmonary vascular remodeling, rather than in affecting pulmonary myogenic tone, hemodynamics, vascular integrity, or vasoconstriction.

We next determined whether pulmonary endothelial CD146 hsa a role in PH pathogenesis. Using 3D-reconstruction post-confocal imaging, we found that, although CD146 was almost undetectable in PAECs, it was expressed in a subset of pulmonary microvascular endothelial cells (PMVECs) lacking pericyte coverage (Supplementary Fig. 11). In lung tissues, endothelial $CD146$ deficiency showed no effects on pulmonary vessel development and integrity (Supplementary Fig. 12). Based on these results, we then studied the development of hypoxic PH using $Cd146^{EC-KO}$ mice. After 4-week hypoxia induction, we detected no changes in lung hemodynamics, RV hypertrophy and small pulmonary arterial remodeling between the two groups (Supplementary Fig. 13). The lack of effects of endothelial CD146 on the development of hypoxic PH might be attributable to the absence of CD146 in PAEC. Moreover, we also found that the responsiveness of HIF-1α and CD146 to hypoxia, as well as the regulation of CD146 and HIF-1α under hypoxia was not robust in PMVECs as well as pericytes (Supplementary Fig. 14), which might account for the inability of endothelial CD146 in hypoxic PH development.

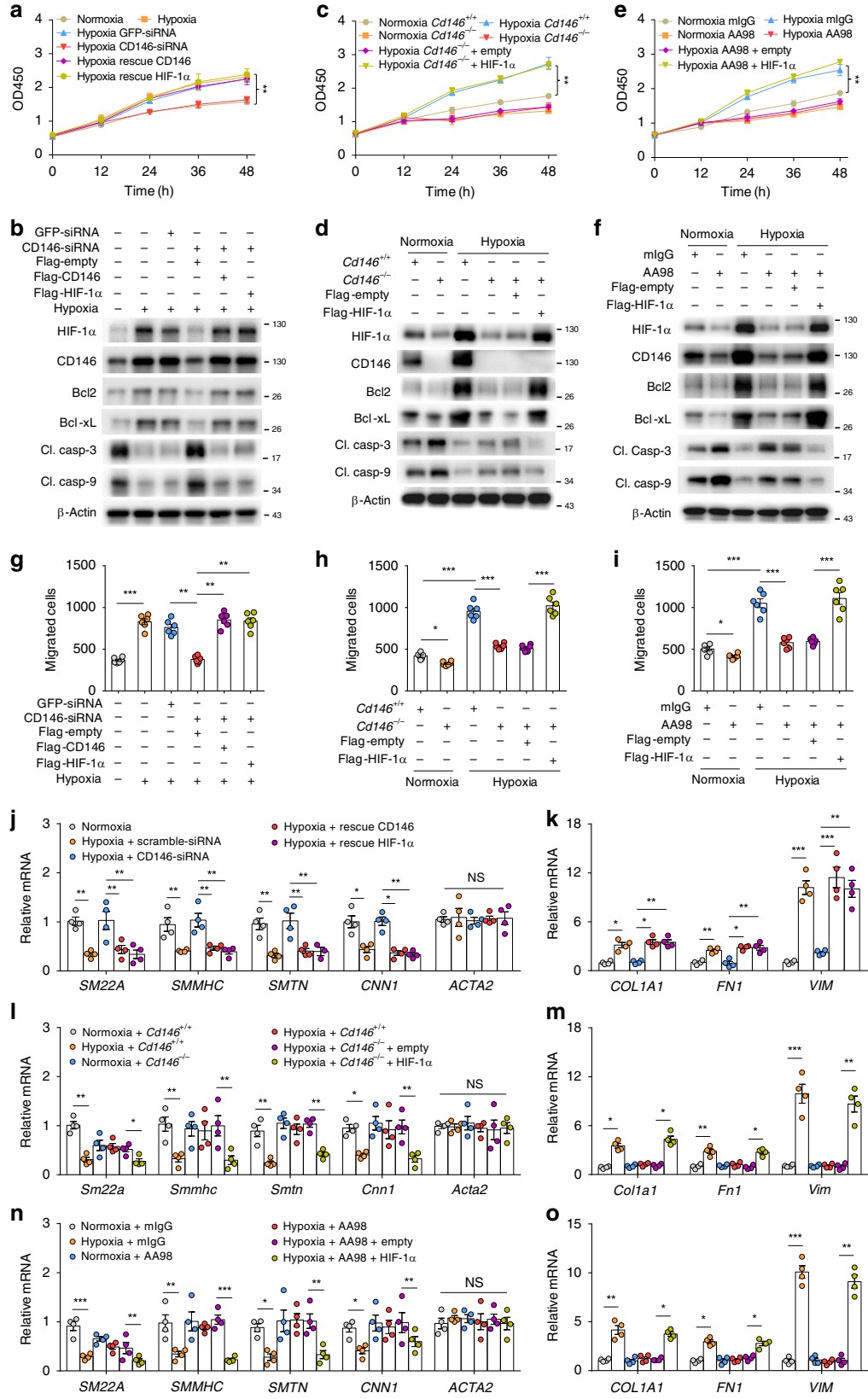

Although there was no difference in hypoxic PH development, we noted a more-impressive phenotype upon an additional 1–month recovery (reoxygenation) following hypoxia (Supplementary Fig. 13a). Compared with WT mice, $Cd146^{EC-KO}$ mice demonstrated persistently elevated RVSP, TPVRI, RV hypertrophy and RV dysfunction after one month in recovery (Supplementary Fig. 13b–j). We propose that the persistent elevation of RVSP and RV failure in $Cd146^{EC-KO}$ mice in recovery might be owing to abnormalities in pulmonary microvascular structure linked to increased vascular resistance and greater cardiac afterload.

**Fig. 4** CD146-HIF-1α axis promotes a synthetic phenotype in PASMCs. **a–d** Human **a**, **b** or mouse **c**, **d** PASMCs were transfected as indicated, and cultured under normoxic or hypoxic conditions, and cell proliferative ability and expression of indicated proteins were determined by CCK-8 assay **a**, **c** or WB **b**, **d**. **e**, **f** Human PASMCs were cultured under normoxic or hypoxic conditions in the presence of anti-CD146 AA98 or control mIgG (50 μg/ml). The cell proliferative ability and the expression of indicated proteins were determined by CCK-8 assay **e** or WB **f**. **g**, **h** Human **g** or mouse **h** PASMCs were transfected as in **a** or **c**. Cell migration was measured in a Transwell Boyden chamber. **i** Human PASMCs were treated as in **e**. Cell migration was measured in a Transwell Boyden chamber. **j–m** Human **j**, **k** or mouse **l**, **m** PASMCs were transfected as in **a** or **c**. The mRNA levels of contractile **j**, **l** and synthetic markers **k**, **m** were detected by real-time RT-PCR. **n**, **o** PASMCs were treated as in **e**. The mRNA levels of contractile **n** and synthetic markers **o** were detected by real-time RT-PCR. $n = 4$ **j–o** or $n = 6$ **a**, **c**, **e**, **g–i** biological replicates for each group. All WB represent data from three **b**, **d**, **f** independent experiments. In all statistical plots, the results are expressed as mean ± s.e.m. *$P < 0.05$, **$P < 0.01$, ***$P < 0.001$. n.s., not significant; by two-tailed Student's $t$ test. Source data are provided as a Source Data file

Indeed, consistent with the hemodynamic data, WT mice demonstrated a significant reduction in muscularization of distal pulmonary arteries upon return to normoxia, whereas no change was noted in $Cd146^{EC-KO}$ mice (Supplementary Fig. 13k–m). Further assessment demonstrated a significant reduction in number of distal vessels in $Cd146^{EC-KO}$ mice after recovery (Supplementary Fig. 13n), suggesting that $Cd146^{EC-KO}$ mice might have a compromised capacity to regenerate pulmonary microvasculature in hypoxia-reoxygenation.

Collectively, these results imply that, the dynamic expression pattern of CD146 in pulmonary vasculature orchestrates PH development at different stages of the disease. Distinct from the role of CD146 in PASMCs that drives PH progression, CD146 in PMVECs might contribute to PH recovery through preservation and proper maintenance of the pulmonary microcirculation. However, the precise mechanisms of endothelial CD146 in PH recovery and its potential contribution to other presentations of PH would require further studies.

**Targeting CD146-HIF-1α axis attenuates experimental PH.** Because CD146-HIF-1α axis contributes to phenotypic switch of PASMCs and vascular remodeling, and targeting CD146-HIF-1α axis with anti-CD146 impaired HIF-1α transcriptional program, we hypothesized that antibody blockade of CD146-HIF-1α effectively prevents the disease. To test this scenario, we induced hypoxic PH in our mouse model, and preventively treated these mice with an intraperitoneal dose of 5 mg kg$^{-1}$ body weight anti-CD146 or control antibody from day 1 to day 28, at 10% oxygen conditions (Supplementary Fig. 15a). As shown in Supplementary Fig. 15b & c, anti-CD146 inhibited the upregulation of CD146 and HIF-1α, as well as HIF-1α target genes. Analogous to the genetic deficiency of $Cd146$, anti-CD146 administration significantly reduced the elevated RVSP, TPVRI, RV/LV + S, pulmonary vascular wall thickness, luminal stenosis, and vascular muscularization, while improving cardiac function in chronic hypoxia-induced PH in mice (Supplementary Fig. 15d–l). Moreover, the anti-remodeling effect of anti-CD146 involved anti-proliferation and pro-apoptosis in the remodeled small pulmonary arteries in anti-CD146-treated mice (Supplementary Fig. 15m & n). However, anti-CD146 treatment under normoxia showed no effect on any of the parameters assessed above. In addition, neither hypoxia- or U46619-induced vasoconstriction was affected by anti-CD146 (Supplementary Fig. 10c & d). These results strongly indicate that preventive targeting of CD146-HIF-1α axis with antibody alleviates progression of hypoxic PH by blocking vascular remodeling.

To further assess whether CD146-HIF-1α axis could serve as a potential therapeutic target for established PH, we used two different PH models: hypoxia-induced PH in mice and MCT-induced PH in rats. We first treated mice that had been exposed to hypoxia for 2 weeks (with apparent elevation of RVSP, Fig. 6d) with anti-CD146 or control antibody for an additional 2 weeks (Fig. 6a). Anti-CD146 treatment inhibited the HIF-1α

transcriptional axis (Fig. 6b, c). Compared with the 2-week group, we found that RVSP, TPVRI, and RV hypertrophy were significantly reduced, without altering SAP (Fig. 6d–g). In addition, antibody treatment enhanced cardiac function as determined by echocardiography (Fig. 6h, i). With respect to vascular remodeling, anti-CD146 markedly reduced pulmonary vascular wall thickness, luminal stenosis, and muscularization (Fig. 6j–m). Moreover, inhibited proliferating PASMCs and increased apoptotic cells (Fig. 6n, o) in the remodeled small PAs were also observed in anti-CD146 treatment group.

We next assessed the efficacy of anti-CD146 treatment in MCT-induced PH in rats (Fig. 7a). In this protocol, anti-CD146 treatment commenced 3 weeks post MCT exposure, a time point at which PH was established. It was found that 2 weeks of anti-CD146 treatment significantly increased lung hemodynamics (Fig. 7b, c), reduced RV hypertrophy (Fig. 7d, e, and Supplementary Fig. 16), improved right heart function (Fig. 7f, g), and suppressed remodeling of the lung vasculature (Fig. 7h–m). Together, these results strongly suggest that targeting of CD146-HIF-1α axis represents a feasible strategy for the treatment of PH by attenuating vascular remodeling.

**Discussion**
In our study, we identified previously unappreciated hypoxic reprogramming involving the cross-regulation between CD146 and HIF-1α as a crucial process in governing the switch of PASMCs toward a more-synthetic phenotype. This CD146-HIF-1α axis-driven phenotypic switch triggers vascular remodeling and fuels the development of PH. On the basis of these observations, we propose that CD146-HIF-1α axis might serve as an attractive therapeutic target for PH.

Despite major progress in the therapy of PAH within the last decade, there is still no cure for this devastating disease. The current treatments for PAH are centred around the pharmacological manipulation of signaling mechanisms used by vasoactive factors, though they exert limited therapeutic benefit owing to their incapacity in blunting vascular remodeling. One possible solution to this challenge is to directly target vascular remodeling to treat PAH. We showed here that this aim is potentially realized by blunting CD146-HIF-1α cross-regulation using anti-CD146, which showed encouraging anti-remodeling effects, even when applied after disease onset. The antibody treatment in both models notably impedes disease progression to a more severe phenotype, without however completely reversing the disease towards a healthy state. Therefore, the proof of concept supports the notion that CD146-HIF-1α axis might be an attractive therapeutic target. In support of this, our in vitro studies show that anti-CD146 potently inhibits hypoxia-induced cell proliferation, migration, the production of synthetic genes, while simultaneously elevating the expression of contractile genes in PASMCs. This might be attributable to the ability of anti-CD146 in blocking transcriptional cascades triggered by HIF-1α, which is a consequence of this antibody to block CD146 dimerization and

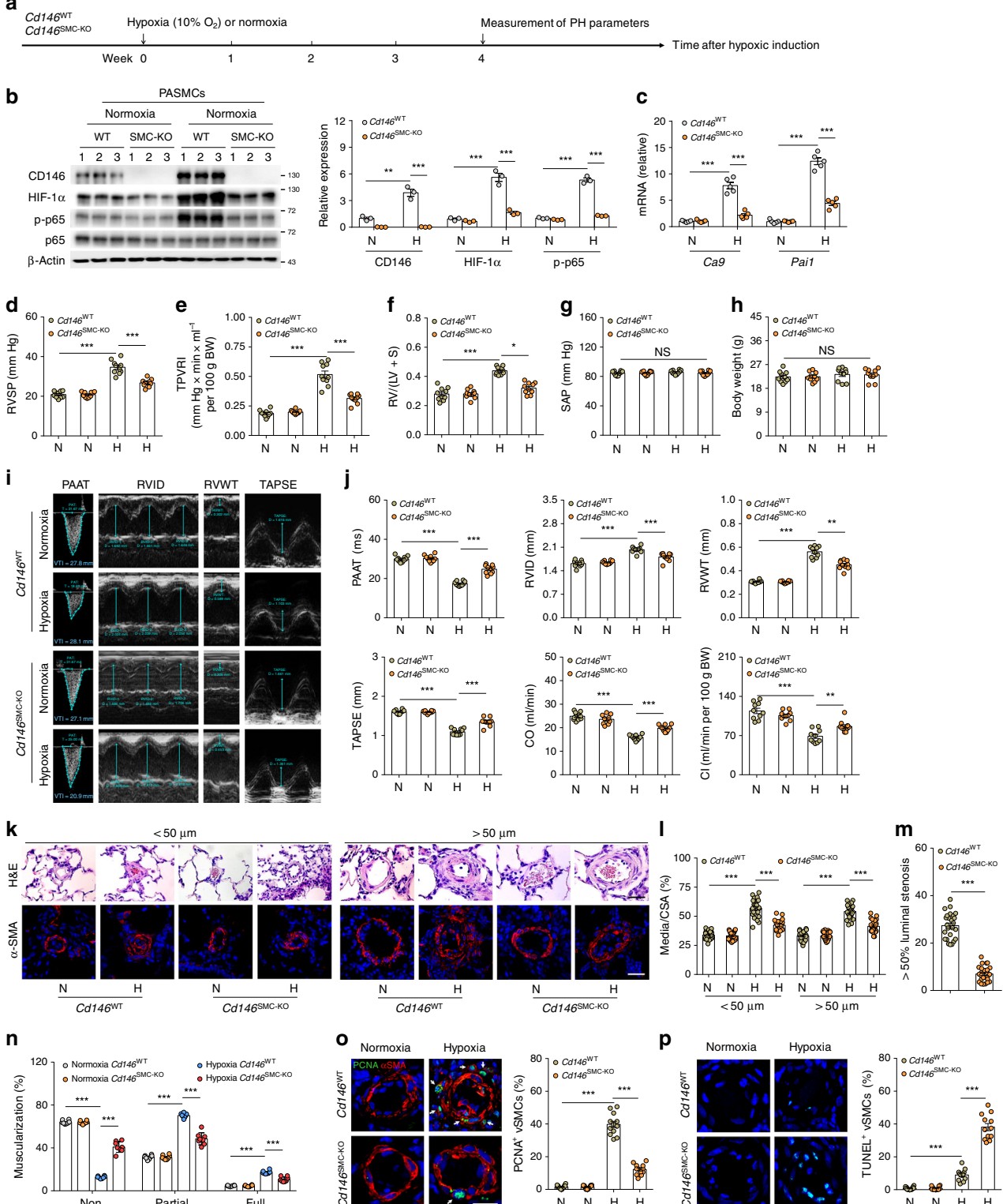

subsequently inhibition of nuclear factor (NF)-kB. Our previous study showed that the structural basis for CD146 dimerization and subsequent NF-kB activation involves domains 4–5 of CD146, an epitope that was specifically recognized by AA98[33]. Concomitantly, ectopic expression of CD146-C452A or CD146-C499A, which could not form dimers, impaired NF-kB activation and HIF-1α hypoxic response. These findings not only provide a feasible strategy for the treatment of PH, but also elucidate the molecular actions of this therapeutic antibody.

The pathogenesis of PAH is complicated and includes an interconnected and reprogrammed network. Therefore, strategies for PAH treatment should target a central mediator that is involved in prominent pathogenic pathways to ultimately inhibit multiple pathological features in a simultaneous manner. CD146-HIF-1α cross-regulation seems to be one such target in the expansion of PASMCs and the exacerbation of PH. It should be noted that HIF-1α hypoxic reprogramming initiates diverging signaling cascades, such as the PDGF-B/PDGFRβ and Notch

**Fig. 5** Disruption of CD146-HIF-1α in SMCs attenuates hypoxic PH. **a** Schematic of hypoxia-induced PH. **b** Left, WB analysis of CD146, HIF-1α and p-p65 in PASMCs from $CD146^{SMC-KO}$ and $CD146^{WT}$ mice. Right, relative expression of CD146, HIF-1α and p-p65 ($n = 3$ biological replicates for each group). **c** The mRNA levels of $Ca9$ and $Pai1$ in PASMCs were measured by real-time RT-PCR ($n = 5$ biological replicates for each group). **d–h** RVSP **d**, TPVRI **e**, SAP **f**, RV/ (LV + S) **g** and body weight **h** in mice after 4 weeks of hypoxia ($n = 10$ mice per group). **i** Echocardiographic (PAAT, RVID, RVWT, and TAPSE) measurements and images. VTI, velocity time integral. **j** Echocardiography measurements were performed on $CD146^{SMC-KO}$ and $CD146^{WT}$ mice after exposure to hypoxic conditions for 4 weeks to determine PAAT, RVID, RVWT, TAPSE, CO, and CI ($n = 10$ mice per group). **k** Representative images of H&E and immunofluorescent staining of PAs (20–50 μm or 51–100 μm in diameter) stained with αSMA (green). Scale bar, 50 μm. **l** Quantification of vascular medial thickness for images in **k** ($n = 5$ mice per group, five PAs per mouse). **m** Quantification of PAs with > 50% luminal stenosis ($n = 5$ mice per group, five PAs per mouse). **n** Proportion of non-, partially-, or fully- muscularized pulmonary arterioles (20–50 μm in diameter) from hypoxia-treated mice ($n = 8$ mice per group). **o** Left, immunofluorescence staining of lung samples for αSMA (red) and PCNA (green). Right, quantification of the relative number of PCNA$^+$/αSMA$^+$ cells. **p** Left, TUNEL staining (green) with DAPI nuclear staining (blue) in small PAs. Right, quantification of the number of TUNEL$^+$ cells. $n = 4$ **o**, **p** mice per group, three PAs per mouse. Scale bar, 50 μm. In all statistical plots, the results are expressed as mean ± s.e.m. *$P < 0.05$, **$P < 0.01$, ***$P < 0.001$. n.s., not significant; by one-way ANOVA with Bonferroni post hoc analysis **d–h**, **j** or two-tailed Student's $t$ test **b**, **c**, **l–p**. All WB represent data from three **b** independent experiments. Source data are provided as a Source Data file

signaling. The hyper-activation of PDGF-B/PDGFRβ signaling has been implicated in the pathobiology of PH[13]. Administration of PDGFRβ inhibitors reversed vascular remodeling in experimental PH[37]. Our previous studies showed that CD146 functions as a co-receptor of PDGFRβ to enable efficient perivascular cell recruitment[18,34]. Thus, these findings provide a second line of evidence that CD146 may be essential to the development of PH by regulating PDGF-B/PDGFRβ signaling after initiating hypoxic reprogramming. Another major pathway, Notch signaling, has also been suggested to play critical roles in PH development[38]. CD146 expression has also been shown to induce Notch signaling in perivascular cells[39]. Therefore, CD146 may affect vascular remodeling through cross-talk with Notch after hypoxic signaling. Together, these observations suggest that CD146 might be at the upstream and convergent point of several signaling nodes: hypoxic signaling, PDGF-B/PDGFRβ signaling, and Notch signaling, all of which have been established in PH.

With regard to anti-CD146 therapy for PH, potential side effects cannot be ruled out, depending on the distribution of CD146 in normal tissues. Previous studies, including ours, showed that CD146 is expressed in a small subset of peripheral T lymphocytes[21,40,41]. Moreover, CD146 is also present in the vascular wall with spatially dynamic localization, including microvacular endothelial cells without perivascular cell coverage, and perivascular cells (SMCs and pericytes) in normal tissues[18,42]. Because CD146 has been implicated in vascular development, cell differentiation, migration, and T-cell activation[43,44], one might expect side effects caused by such anti-CD146 therapy. We previously reported that AA98 fails to affect proliferation, differentiation and activation of lymphocyte as well as other CD146$^+$ immune cells[21,41], suggesting that AA98 might not elicit adverse effects by sparing the host's immune response. Regarding vascular wall, we previously screened 48 normal tissues and observed that AA98 recognized blood vessels with low frequency. In the few cases where AA98 recognized vessels, only perivascular cells were stained[42]. Although AA98 shows a low propensity to recognize normal blood vessels, we cannot exclude possible side effects of this antibody treatment by targeting vascular wall. This safety issue needs to be addressed in future studies.

In addition to potential side effects, effectiveness of the antibody treatment must be verified before translating this therapy for human PAH. In this study, we provide convincing evidence that CD146 is elevated in small PAs from human PAH lungs; anti-CD146 impeded the switch of human PASMCs towards a more synthetic phenotype. These results provide a rationale for the potential application of this antibody-based therapy for human PAH. It is tempting to postulate that anti-CD146 antibody might be efficacious across a spectrum of PH etiologies, e.g., idiopathic pulmonary artery hypertension (IPAH) and PH

associated with hypoxemia, as SMC expansion represents a common pathological hallmark of the disease. Future efforts to determine the effectiveness of anti-CD146 treatment, as well as understanding potential side effects, should lead to promising therapeutics for human PAH.

In this study, we provide compelling evidence that under hypoxia, the expression of CD146 and HIF-1α in PASMCs is reciprocally regulated. We found that hypoxic or hypoxia-mimicking conditions promote $CD146$ transcription in a time- and $O_2$ concentration-dependent manner. In addition, $CD146$ is specifically upregulated by HIF-1α, but not by HIF-2α and TP53, despite that the latter two are also well-known hypoxic transcriptional factors[45,46]. Moreover, this regulation was further shown to occur through direct binding of HIF-1α to the specific HRE within $CD146$ promoter. Interestingly, PASMCs in culture also express low levels of HIF-1α. Therefore, the transcriptional activation of $CD146$ by HIF-1α was also observed under basal conditions. Previous studies have established that HIFs are expressed in a cell-type specific manner in the lung, for instance, HIF-1α is ubiquitously expressed, whereas HIF-2α is restricted to the endothelium and type II pneumocytes[47,48]. These studies agree with our finding that CD146 expression in PASMC is regulated by HIF-1α. In turn, CD146 promotes $HIF1A$ transcription and activity, thus forming a positive feedback loop to ensure HIF-1α transcriptional cascades. These results are consistent with previous studies that CD146 expression was positively correlated with nuclear HIF-1α[24,25], and CD146-positive cells were more responsive to hypoxia[26]. Our study identified an important upstream regulator for cellular hypoxic response and implicates the existence of a mutual regulatory loop between CD146 and HIF-1α.

Previous findings identified a connection between HIF-1α and PAH pathogenesis, but it is unclear how hypoxia activates HIF-1α in PASMCs. Although early studies demonstrated the induction of $Hif1a$ mRNA in experimental animals during hypoxia[49,50], numerous in vitro studies led to the current model that HIF-1α is regulated predominantly at the post-translational level through the inhibition of $O_2$-dependent PHDs that drive HIF-1α degradation under hypoxia[51,52]. Here, we clearly show that in PASMCs, NF-kB serves as a hypoxia-regulated transcriptional factor that maintains $HIF1A$ mRNA expression. CD146 triggers HIF-1α transcription by activating NF-kB. This process precedes HIF-1α protein accumulation. Thus, our study has uncovered a previously unidentified feedback loop orchestrated by CD146, NF-kB, and HIF-1α that ensures hypoxic reprogramming in PASMCs. In addition, the ability of NF-kB to promote HIF-1α activation linked by CD146 might have far-reaching physiological implications in PH, because it indicates the presence of a coupling mechanism between two evolutionary ancient stress responses:

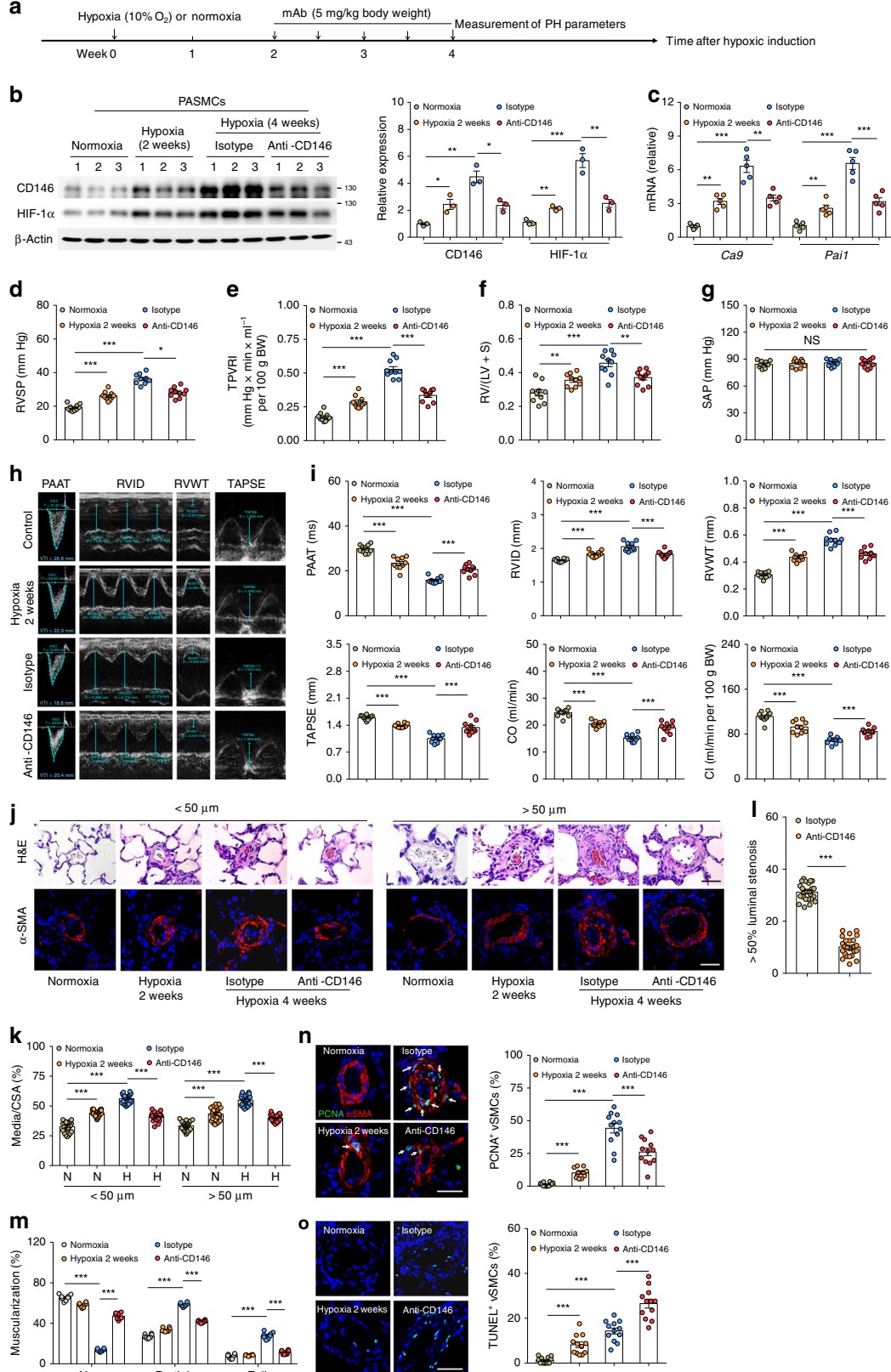

the inflammatory response and the hypoxic response. By controlling HIF-1α activation in PASMCs under hypoxia or possibly during chronic lung diseases, which may lower local O₂ tension, CD146-mediated NF-kB expands its well-established proinflammatory function, because HIF-1α-dependent hypoxic response is critical for providing PASMCs and maybe other types of pulmonary cells undergoing hypoxia with sufficient energy supplies and allows them to resist cell death and apoptosis. Therefore, CD146-mediated coupling of the two ancient stress responses establishes a proinflammatory and pro-survival

**Fig. 6** Targeting CD146-HIF-1α axis attenuates hypoxic PH in mice. **a** Schematic of CD146-targeted therapy in hypoxia-induced PH in mice. **b** Left, WB analysis of CD146 and HIF-1α in PASMCs isolated from anti-CD146- or mIgG-treated mice. Right, relative expression of CD146 and HIF-1α ($n = 3$ biological replicates for each group). **c** mRNA levels of *Ca9* and *Pai1* in PASMCs were detected by real-time RT-PCR ($n = 5$ biological replicates for each group). **d**–**g** RVSP **d**, TPVRI **e**, SAP **f** and RV/(LV + S) **g** in anti-CD146- or mIgG-treated mice after 4 weeks of hypoxia ($n = 10$ mice per group). **h** Echocardiographic (PAAT, RVID, RVWT, and TAPSE) measurements and images. **i** Echocardiography measurements were performed on anti-CD146- or mIgG-treated mice to determine PAAT, RVID, RVWT, TAPSE, CO and CI ($n = 10$ mice per group). **j** Representative images of H&E and immunofluorescent staining of PAs (20–50 μm or 51–100 μm in diameter) stained with αSMA (green). Scale bar, 50 μm. **k** Quantification of vascular medial thickness for images in **j** ($n = 5$ mice per group, five PAs per mouse). **l** Quantification of PAs with >50% luminal stenosis ($n = 5$ mice per group, five PAs per mouse). **m** Proportion of non-, partially-, or fully muscularized pulmonary arterioles (20–50 μm in diameter) from hypoxia-treated mice ($n = 8$ mice per group). **n** Left, immunofluorescence staining of lung samples for αSMA (red) and PCNA (green). Right, quantification of the relative number of PCNA⁺/αSMA⁺ cells. **o** Left, TUNEL staining (green) with DAPI nuclear staining (blue) in small PAs. Right, quantification of the number of TUNEL⁺ cells. $n = 4$ **n**, **o** mice per group, three PAs per mouse. Scale bar, 50 μm. All WB represent data from three **b** independent experiments. In all statistical plots, the results are expressed as mean ± s.e.m. \*$P < 0.05$, \*\*$P < 0.01$, \*\*\*$P < 0.001$. n.s., not significant; by one-way ANOVA with Bonferroni post hoc analysis **d**–**g**, **i** or two-tailed Student's $t$ test **b**, **c**, **k**–**o**. Source data are provided as a Source Data file

microenvironment to permit vascular remodeling and PH exacerbation.

In conclusion, our study reveals a causative role of CD146-HIF-1α axis in pulmonary vascular remodeling. From a clinical perspective, our findings should greatly facilitate the development of potential anti-remodeling therapies for PH, and perhaps other vascular remodeling disorders.

## Methods

**Antibodies, reagents, and plasmids.** Anti-CD146 monoclonal antibodies, including AA98 and AA4, were generated and characterized in our laboratory[20,21]. AA98 was used for functional assays; AA4 (1:50 dilution) for immunohistochemistry (paraffin-embedded); rat anti-mouse CD146 (1:50 dilution for immunohistochemistry, 1:1000 dilution for immunoblots, clone: ME-9F1, Biolegend) and rabbit anti-CD146 (1:50 dilution for immunohistochemistry, 1:1000 dilution for immunoblots, ab75769, abcam) for immunohistochemistry (frozen section) and immunoblots. Other antibodies were used in this study: NF-κB p65 (1:2000 dilution, 8242, Cell Signaling), p-p65 (1:1000 dilution, 3031, Cell Signaling), Bcl2 (1:2000 dilution, 3498, Cell Signaling), Bcl-xL (1:2000 dilution, 2764, Cell Signaling), cleaved Caspase 3 (1:1000 dilution, 9664, Cell Signaling), cleaved Caspase 9 (1:1000 dilution, 7237, Cell Signaling); HIF-1α (1:1000 dilution, ab1, abcam), SMMHC (1:50 dilution for immunohistochemistry, 1:1000 dilution for immunoblots, ab53219, abcam), Desmin (1:1000 dilution, ab15200, abcam), αSMA (1:50 dilution for immunohistochemistry, 1:1000 dilution for immunoblots, ab5694, ab21027, abcam), NG-2 (1:1000 dilution ab5320, abcam), CD31 (1:500 dilution, ab56299, ab24590, abcam), Flag (1:5000 dilution, ab49763, abcam), Myc (1:5000 dilution, ab32, abcam), HIF-2α (1:1000 dilution, ab199, abcam), TP53 (1:1000 dilution, ab28, abcam), β-actin (1:5000 dilution, AM1021B, Abgent), PCNA (1:50 dilution, sc-7907, Santa Cruz), isotype-matched control antibody mIgG (M5409, Sigma-Aldrich).

CoCl₂, dimethyloxallyl glycine (DMOG), and desferrioxamine (DFX) were purchased from Santa Cruz. Digoxin and acriflavine were obtained from Sigma-Aldrich. U46619 (56985-40-1) and actinomycin D (A4448) were purchased from APExBIO. MCT (HY-N0750) was purchased from MedChemExpress (MCE).

On the basis of CD146 cDNA provided by Dr. Judith P. Johnson (University of Munich, Germany), DNA fragments encoding full length CD146 was cloned into p3XFLAG-CMV™-14 expression vector[18,23]. Flag-HIF-1α was purchased from Sino Biological Inc. Luciferase construct containing the human wild-type *CD146* promoter 1 kb sequence upstream of translational start site was cloned into a pGL3 luciferase construct using *Kpn*I and *Bgl*III as insertion sites. The corresponding HRE mutant construct (a substitution of the 5′-GCGTGG-3′ motif by 5′-GGCAGG-3′) was generated by site-directed mutagenesis. A luciferase construct containing five copies of the HRE sequence[53], or carbonic anhydrase 9 promoter was cloned into a pGL3 luciferase vector[54]. A luciferase construct containing *HIF1A* promoter (−538 to +284) was cloned into a pGL3 luciferase vector. The corresponding mutant construct for NF-κB binding site (−197 to −188) was generated by site-directed mutagenesis[55]. The fidelity and appropriate mutations of all the constructs generated were confirmed by sequencing.

**Animals.** C57BL/6 J mice and Sprague Dawley rats were obtained from the Beijing Vital River Laboratory Animal Technology Co., Ltd. and the department of Laboratory Animal Science, Peking University Health Science Center. All animals were maintained in a pathogen-free facility. All animal experiments were performed in compliance with the guidelines for the care and use of laboratory animals and were approved by the institutional biomedical research ethics committee of the Institute of Biophysics, Chinese Academy of Sciences.

Previous studies have established that PDGFRβ is strongly expressed in the concentric layers of SMCs of developing PAs, and is critically involved in inducing

the medial layer during the formation of pulmonary arterial wall[56]. Moreover, the migration and clonal expansion of the hypoxia-primed PDGFRβ⁺ SMCs give rise to the pathological smooth muscle coating of small pulmonary arterioles[36]. Therefore, *Pdgfrβ*cre mouse strain (obtained from Beijing Biocytogen Co., Ltd.) was used for generating SMC-specific *Cd146* knockout mouse (*Cd146*SMC-KO)[18]. In brief, for generating *Cd146*SMC-KO mice, *Pdgfrβ*cre mice were first crossed with *Cd146*floxed/floxed mice. The *Pdgfrβ*cre*Cd146*floxed/+ mice were then backcrossed with *Cd146*floxed/floxed mice to obtain *Pdgfrβ*cre*Cd146*floxed/floxed mice, which were annotated *Cd146*SMC-KO mice. *Cd146*floxed/floxed mice (which were annotated as *Cd146*WT mice here) were used as controls. All genotypes were confirmed by PCR analysis and identified by sequencing. *Cd146* deficiency in lung SMCs was confirmed using western blotting (WB) analysis (Supplementary Fig. 2). *Tie2*cre mouse strain was used for generating endothelial-specific *Cd146* knockout mouse (*Cd146*EC-KO) using similar strategies with that of *Cd146*SMC-KO.

**PH models in rodents and antibody-based therapy.** For chronic hypoxia-induced PH model in mice, 8- to 10-week-old male *Cd146*SMC-KO or *Cd146*EC-KO mice and their age- and weight-matched WT littermates were exposed to room air (normoxia) or 10% oxygen (hypoxia) in a ventilated chamber (BioSpherix). All animals had access to standard mouse chow and water, and they were subjected to an alternating 12-hour light/dark cycle under controlled temperature conditions. The chamber was opened twice a week for 10 min for cleaning and replenishing food and water supplies. For preventive antibody treatment, C57BL/6 J mice were intraperitoneally injected with the anti-CD146 mAb AA98 or the control mouse IgG (5 mg per kg body weight, twice a week) from the day of hypoxic induction. For therapeutic antibody treatment experiments in C57BL/6 J mice, the antibodies were injected 2 weeks after hypoxia induction (with antibody 5 mg per kg body weight, twice a week). Echocardiography, hemodynamics, and RV hypertrophy were assessed (as described below) 28 d after hypoxia induction.

For generating MCT-induced PH, Sprague Dawley rats (male, ~200–250 g) were intraperitoneally injected with 50 mg per kg body weight MCT, allowed to develop PH from days 0 to 21. Control rats were given i.p. injections of sterile water as a vehicle control. After 21 day MCT induction, rats were treated with anti-CD146 mAb AA98 or the control mIgG (5 mg per kg body weight, twice a week) from day 21 to killing on day 35. Echocardiography, hemodynamics, and RV hypertrophy were assessed 35 d after MCT injection.

**Echocardiography.** Echocardiography was performed using the Vevo 2100 high-resolution imaging system (FUJIFILM VisualSonics, Toronto, Canada) equipped with a 18- to 38-MHz (MS400, mouse cardiovascular) or a 13- to 24-MHz (MS250, rat cardiovascular) scan head. Rodents were placed on a homeothermic plate at a supine position to minimize heat loss. Rectal temperature, heart rate and respiratory rate were recorded continuously throughout the study. For anesthesia, rodents were placed in an anesthesia induction chamber, and isoflurane was delivered using a vaporizer at 0.5–1.5%. The rodents were depilated and preheated ultrasound gel was applied. The heart rates were maintained at 450–500 (mice) and 325–350 (rats) beats per minute (bpm), whenever possible. Transthoracic echocardiography was performed to measure right ventricle internal diameter (RVID), TAPSE, PAAT, and CO as described below[57,58].

In brief, right ventricle parameters, including RVID and RVWT at the end of a diastole were determined using M-mode from the right parasternal short axis view. To determine TAPSE, the M-mode cursor was oriented to the junction of the tricuspid valve plane and the RV free wall using the apical four-chamber view. PA diameters were measured at the level of the pulmonary outflow tract during midsystole using the left ventricular long-axis view. PAAT and the PA flow VTI were measured using Pulsed-wave Doppler. PAAT was defined as the interval from the onset to the maximal velocity of forward flow, and was measured from the pulsed-wave Doppler flow velocity profile of the RV outflow tract in the left ventricular long-axis view. By combining PA VTI, pulmonary artery area and heart rate, the

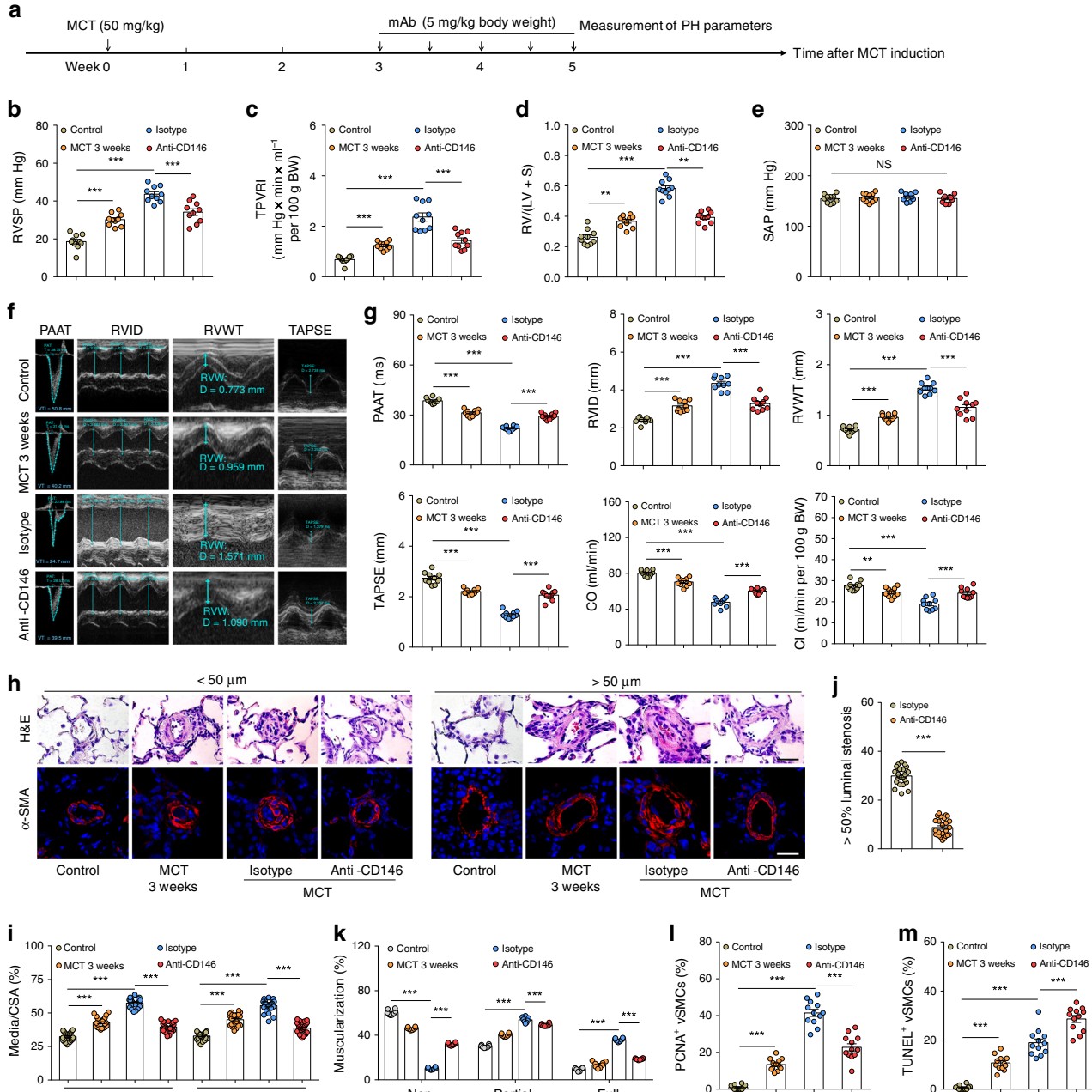

**Fig. 7** Targeting CD146-HIF-1α axis attenuates MCT-induced PH in rats. **a** Schematic of CD146-targeted therapy in MCT-induced PH in rats. **b–e** RVSP **b**, TPVRI **c**, SAP **d** and RV/(LV + S) **e** in anti-CD146- or mIgG-treated rats after 5 weeks of MCT (*n* = 10 rats per group). **f** Echocardiographic (PAAT, RVID, RVWT, and TAPSE) measurements and images. **g** Echocardiography measurements were carried out on anti-CD146- or mIgG-treated rats after 5 weeks of MCT to determine PAAT, RVID, RVWT, TAPSE, CO, and CI (*n* = 10 rats per group). **h** Representative images of H&E and immunofluorescent staining of PAs (20–50 μm or 51–100 μm in diameter) stained with αSMA (green). Scale bar, 50 μm. **i** Quantification of vascular medial thickness for images in **h** (*n* = 5 rats per group, five PAs per rat). **j** Quantification of PAs with >50% luminal stenosis (*n* = 5 rats per group, five PAs per rat). **k** Proportion of non-, partially-, or fully- muscularized pulmonary arterioles (20–50 μm in diameter) from MCT-treated rats (*n* = 8 rats per group). **l** Quantification of the relative number of PCNA+/αSMA+ cells (*n* = 4 rats per group, three PAs per rat). **m** Quantification of the number of TUNEL+ cells (*n* = 4 rats per group, three PAs per rat). In all statistical plots, the results are expressed as mean ± s.e.m. *P < 0.05, **P < 0.01, ***P < 0.001. n.s., not significant; one-way ANOVA with Bonferroni post hoc analysis **b–e**, **g** or two-tailed Student's *t* test **i–m**. Source data are provided as a Source Data file

echocardiographically derived CO was determined. Total pulmonary vascular resistance (TPVRI) was calculated using the following formula: TPVRI = RVSP/CI, where RVSP is right ventricular systolic pressure (mm Hg), and the CI (ml min⁻¹ per 100 g body weight) is the CO (ml min⁻¹) normalized to 100 g body weight.

**Hemodynamic measurement**. The measurement of the right ventricular pressure was based on the direct catheterization of the right ventricle via the jugular vein in close-chested rodents[59]. Anesthesia was given to rats or mice as described above.

After incision of the skin starting from the mandibles and extending to the sternum, the thyroid grand was carefully dissected to expose the right jugular vein. Right heart catheterization was performed by inserting a rigid cannula containing a flexible 0.9-Fr micro-tip pressure transducer via the right jugular vein into the right ventricle to assess RVSP. Subsequently, the catheter was inserted into the aorta and the LV through the left carotid artery for measurement of SAP. Data were collected and analyzed using the PowerLab data acquisition system (AD Instruments) and LabChart 7.2 software.

**Histological analysis**. After measuring all hemodynamic parameters, the pulmonary circulation was flushed with chilled PBS, and the heart and lungs were removed. The right ventricle (RV) was carefully dissected from the heart and weighed. Right ventricular hypertrophy was assessed by normalizing the weight of the RV to the weight of the left ventricle plus septum (RV/LV + S).

The left lungs were placed in liquid nitrogen for preparation of homogenates, and the lower lobes of the right lungs were fixed for 24 h in 4% paraformaldehyde solution. After paraffin embedding and sectioning, the slides (5-μm thickness) were stained with hematoxylin and eosin) for morphological analysis. Pulmonary vascular remodeling was quantified by accessing the medial wall thickness and the percentage of muscularization. To determine the degree of medial wall thickness, 20–25 muscular arteries categorized 20–50 μm and 51–100 μm in diameter from each lung were randomly outlined by an observer blinded to mouse genotype or antibody treatment. The degree of medial wall thickness, expressed as a ratio of medial area to cross sectional area (i.e., Media/CSA)[60], were analyzed using image J. To access the degree of muscularization, 40–60 intraacinar vessels at a size between 20 and 50 μm in each mouse were categorized as nonmuscular (i.e., no apparent muscle), partially muscular (i.e., with only a crescent of muscle), or muscular (i.e., with a complete medial coat of muscle), as previously reported[37]. The degree of muscularization was expressed as ratio of non-, partially-, fully muscular vessels to the number of total vessels.

**Immunofluorescence and in vivo apoptosis and proliferation**. Human lung sections (Paraffin-embedded) from three patients with IPAH were obtained from Fuwai Hospital. Lung sections from three control donors with no cardiac or pulmonary diseases were obtained from the Alenabio Biotechnology Co. Ltd (Xi'an, China) and Tongxuxian Renmin Hospital. The protocol for human research was approved by the Ethics Committees of Fuwai Hospital and Tongxuxian Renmin Hospital. Experiments were performed in accordance with the ethical regulations of the mentioned committees. Written informed consent was obtained from all patients or their next of kin.

Paraffin-embedded lung tissue sections (5-μm thick) were deparaffinized in xylene and rehydrated in a graded ethanol series to PBS (pH 7.2). Antigen retrieval was performed by pressure cooking of lung sections in citrate buffer (pH 6.0) for 15 min. Double immunofluorescence staining of lung sections was performed using primary antibodies to CD146 as well as either αSMA or CD31. After overnight incubation, slides were washed and incubated with the respective secondary antibodies, Alexa 488- and Alexa 555-conjugated secondary antibodies for 1 h. All sections were counterstained with nuclear 4′,6-diamidino-2-phenylindole (DAPI) and mounted with fluorescent mounting medium. Images were taken using a confocal laser scanning microscope (Olympus FLUOVIEW FV 1000) with an Olympus IX81 digital camera. To detect proliferation of PASMCs, the lung tissue sections were also stained for PCNA expression. To assess apoptosis, sections were visualized using a TUNEL method, with a commercially available in situ cell death detection kit (11 684 795 910, Roche Molecular Biochemicals, Mannheim, Germany) and were counterstained with nuclear DAPI.

**Analysis of small pulmonary arterial vasoconstriction**. The hypoxia- or drug-induced vasoconstriction of small PAs was determined by videomorphometric analysis of cross-sectioned arteries in precision cut lung slices[61]. In brief, the lungs were filled with low melting point agarose and cut into 200 μm thick precision cut lung slices (PCLS) using vibratome. After removing the agarose by bubbling the MEM with the normoxic gas for 2 h at 37 °C, one PCLS was transferred into the flow-through superfusion chamber in which it was exposed to hypoxic medium (gassed with 1% O₂) or normoxic medium supplemented with U46619 (0.1 μM; no flow) in the presence or absence of anti-CD146 antibody (50 μg/ml). The intraacinar PAs (located at gussets of alveolar septa next to alveolar ducts) with inner diameters of 20–40 μm were selected for videomorphometric analysis of vasoconstriction. The luminal area of the vessels was calculated using Image J software. The area of the vessel lumen at the beginning of the experiment was defined as 1. Vasoreactivity was recorded as changes in the luminal area.

**RNA extraction and real-time PCR**. Total RNA from lung homogenates of mouse or rat, PASMCs from human and mouse were extracted using Trizol reagent (Invitrogen), according to the manufacturer's instructions. In brief, total RNA (2 μg) was reverse-transcribed into cDNA using Reverse Transcription Reagent kits (Transgene), according to the manufacturer's instructions. Real-time quantitative PCR (qPCR) analysis was performed on an Applied Biosystems StepOnePlus real-time PCR instrument (ABI 7500) in combination with a SYBR Green PCR mix (Toyobo Co., Osaka, Japan). All samples were amplified using biological triplicates with two technical replicates per sample. The 7500 Sequence Detection System software (Applied Biosystems) was used to obtain CT values. Results were analysed using the comparative CT method. Samples were normalized to β-actin to account for cDNA loading differences. The primer sequences for PCR are summarized in Supplementary Table 1 and 2.

**PASMC isolation and culture**. Primary murine PASMCs were isolated from Cd146^SMC-KO and Cd146^WT mice using a modified elastase/collagenase digestion protocol[57]. In brief, pulmonary arteries were first isolated from the left lobe of lung

tissues owing to the relative ease of working with a single, large lobe[62]. The intrapulmonary arteries of the first to third order branches were selected and carefully dissected away from connective tissues under a light microscope. After removing the adventitia, the isolated pulmonary arteries were digested in dispersion medium containing 40 μmol/l CaCl₂, 1 mg/ml elastase, 1 mg/ml collagenase, 0.2 mg/ml soybean trypsin inhibitor, and 2 mg/ml albumin for 30 min at 37 °C. After filtration with 100-μm cell strainers, PASMCs were collected through centrifugation at 225 × g for 10 min and subcultured with Dulbecco's Modified Eagle's medium (DMEM) containing 10% fetal bovine serum (FBS) and antibiotics. After the primary cells grew to 90% confluence, the cells were digested, and the identity and Cd146 deficiency were confirmed by WB analysis with SMC-specific markers (Supplementary Fig. 2). Only early-passage (passage 3–4) PASMCs were used for all experiments.

Human PASMCs were purchased from ScienCell Research Laboratories (ScienCell, 3110). 293 T cells were obtained from ATCC (ATCC, CRL-3216). For hypoxia exposure expriments in vitro, PASMCs seeded in culture dishes were placed inside a hermetic tank that contained 1–21% O₂/5% CO₂. Cells were routinely tested for mycoplasma contamination.

**PAEC isolation and culture**. Primary PAECs were isolated from mice according to protocols with slight modifications[18,62]. Mice 8–10 weeks of age were killed and lungs were extracted. Pulmonary arteries were carefully isolated from mice and the vascular tubes were washed with DMEM and centrifuged at 205 × g for 5 min. The arteries were resuspended in 1 ml/lung DMEM and digested with 200 mg/ml dispase and 0.1%/1 mg/ml collagenase II for 1 h at 37 °C. After separating the enzyme localized in the arteries by centrifugation and repeated washing in DMEM, cell suspensions were plated onto 6-well plates coated with rat tail collagen type I. EBM2 media was used to culture the cells. After the primary cells reached 90% confluency, they were digested and sorted by fluorescence-activated cell sorting according to their anti-CD31 staining patterns for further experimental procedures. The primary isolated PAECs were immediately used for immunoblot experiments.

**Gene interference**. Small interfering RNA (siRNA) targeting human CD146, HIF1A, HIF2A, TP53, and p65 was transiently transfected into PASMCs grown to ~80% confluency with Lipofectamine RNAiMAX (Invitrogen Life Technologies), according to the manufacturer's instructions. A scrambled siRNA was used as a negative control. Sequences of siRNA are shown in Supplementary Table 3. For delivery of CD146- or HIF-1α-expressing plasmids into PASMCs, we used SMCFectagen reagent (Science Cell) which was specifically optimized for efficient transfection of primary smooth muscle cells. Knockdown or over-expression efficiency was determined by WB or real-time PCR 36 h after transfection.

**Cell migration assay**. Cell migration experiments were performed using a 96-well Boyden chamber (Corning Costar) containing a filter with a pore size of 8 μm. Human PASMCs or primary mouse PASMCs (Cd146^+/+ or Cd146^−/−) were transfected as indicated and incubated in 21% or 5% O₂ for 48 h. The same number (5 × 10³) of cells were grown with DMEM (with 5% FBS) in the upper chamber. The cell migration was initiated by filling the lower chamber with DMEM (with 10% FBS) and incubation for 12 h. The cells that migrated to the lower membrane was stained with crystal violet and counted under a microscope.

**Cell proliferation assay**. Cell proliferation analysis was performed using the cell counting kit-8 (CCK-8) (Beyotime) according to the manufacturer's instructions. In brief, the same number (5 × 10³) of human PASMCs or primary mouse PASMCs (Cd146^+/+ or Cd146^−/−) cultured in DMEM (with 10% FBS) were transfected and incubated in 21% or 5% O₂ for time periods indicated. CCK-8 solution (10 μl) was added to each well of the plate and then incubated for 3 h in the incubator for measurement of absorbance at 450 nm using a microplate reader.

**Dural luciferase reporter assay**. The PASMCs were transfected as indicated and exposed to hypoxia for 24 or 36 h. The pGL3-Basic vectors containing different promoters were transfected into PASMCs, together with pRLTK containing the Renilla luciferase reporter gene and the control empty plasmid. Twelve hours after transfection, Firefly and Renilla luciferase activities were then measured with the Dual-Luc Assay Kit (Promega).

**ChIP and PCR**. Specific protein–DNA interactions were examined by ChIP followed by qPCR (Chromatin Immunoprecipitation Assay Kit, Millipore). Protein–DNA cross-linking was performed by fixating cells in 1% formaldehyde for 10 min at room temperature. DNA-protein complexes from 2 × 10⁶ cells were sheared to lengths between 200 and 500 base pairs by sonicator. The pre-cleared fragments were incubated with 10 μg of HIF-1α or p65 specific antibody, or IgG (as a negative control) overnight, followed by immunoprecipitation by Protein A. Cross-linking was reversed by heating at 65 °C overnight, followed by Proteinase K digestion at 45 °C for 2 h. DNA was then recovered with QIAquick PCR purification kit (Qiagen) for qPCR to prove affinity against CD146 or HIF1A promoter region. Quantitative PCR was performed with primers for the CD146 promoter (forward: 5′-ttctgggagacaggctgtag-3′; reverse: 5′-agggtaagagggacttgcaag-3′) flanking

the HIF-1α binding site (−692/−688 bp), and with primers for the *HIF1A* promoter (forward: 5′-gaacagagagcccagcagag-3′; reverse: 5′-tgtgcactgaggagctgagg-3′) flanking the NF-κB binding site (−197/−188 bp).

**Western blotting**. For immunoblotting experiments, proteins were separated by 10% sodium dodecyl sulphate-polyacrylamide gel electrophoresis and then transferred onto a nitrocellulose membrane. Membranes were then blocked with 5% non-fat milk in phosphate-buffered saline with 0.1% Tween-20 for 1 h, incubated for 1 h with primary antibodies and then probed with horseradish peroxidase-conjugated anti-mouse, anti-rat, or anti-rabbit secondary antibodies. All immunoblots were carried out using chemiluminescence reagent (Pierce) and the signals were collected by ChemiScope3600MINI (Clinx Science Instruments). Densitometry was performed using Gel analysis software. Uncropped versions of western blots performed can be found in the Source data file.

**Lung water content**. In brief, mice were killed and lungs were extracted. The wet weight of lung tissues was determined immediately following extraction. After drying for 24 h at 100 °C, the dry weight of lungs was measured. Percentage water content was determined using the following formula: water content (%) = (wet weight−dry weight) × 100/wet weight.

**Tracers' permeability**. Mice at 8–10 weeks were given an intravenous injection of Evan's blue dye, respectively. The mice were killed and perfused with PBS through the left ventricle. Lungs were then extracted and dissolved in formamide. After extraction in formamide for 48 h at 55 °C, the absorption (620 nm) of the Evan's blue extracted from the lung was measured using a microplate spectrophotometer.

Pulmonary vascular permeability was also determined using a FITC-Dextran extravasation technique. Mice were intravenously injected with FITC-Dextran (4 kD or 70 kD) at 1 mg per 20 g body weight, followed by euthanasia 30 min later. Lungs were perfused with PBS, dissected, and homogenized for fluorescence quantification of the tracer. Lung homogenates were prepared in radioimmunoprecipitation buffer supplemented with protease inhibitors and phosphatase inhibitors, briefly sonicated and quantified using a protein assay kit (Bio-Rad) according to the manufacturer's manual. Fluorescence was measured in lung homogenates using a Fluoroskan microplate reader (excitation, 485 nm; emission, 530 nm).

**Statistical analysis**. All experiments were performed independently for two or three times. All summary data are expressed as mean ± s.e.m. (standard error of the mean). One-way analysis of variance test with Bonferroni post hoc analysis or two-tailed Student's *t* test was used to compare differences between groups in various experiments. Differences of $P < 0.05$ were considered to be statistically significant.

**Reporting summary**. Further information on research design is available in the Nature Research Reporting Summary linked to this article.

## Data availability

Source data for all graphs and full scans of the blots in the figures of this manuscript are provided in Source Data file. All data are available from the corresponding authors upon reasonable request.

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

## Acknowledgements

We thank all members of the Yan XY laboratory for critical discussions of our manuscript. We also thank Dr. Torsten Juelich (Peking University) for careful reading and editing of our manuscript. We are grateful to Dr. Qing Xu (Core Facilities Centre, Capital Medical University) for assisting in echocardiography and hemodynamic experiments. We thank the Center for Experimental Animals (China Agricultural University and Institute of Biophysics) for the assistance in PH model induction and hemodynamic measurements. We are grateful to Drs. Junfeng Hao and Yan Teng for technical support in histological and immunofluorescent analysis. This work was supported in part by grants from the National Basic Research Program of China (2015CB553705) and the National Key R&D Program of China (2017YFA0205503).

## Author contributions

Y.L. conceived the study, designed and performed experiments, analyzed data, prepared the figures, and wrote the manuscript; X.T. and L.Z. helped in animal experiments and provided human PASMCs; J.C. conducted lung permeability experiments and constructed the genetically engineered mice; Z.L. and X.C. prepared anti-CD146 monoclonal antibodies; S.Z. and S.Y. helped in PH model induction and animal experiments; J.F. handled the funding; and X.Y. supervised the study, reviewed the data and wrote the manuscript.

## Additional information

**Competing interests:** The authors declare no competing interests.

