## [Peer Review File · Nature Communications]

Reviewers' comments:

Reviewer #1 (Remarks to the Author):

The present study uncovered a previously unrecognized cross-regulation between the adhesion molecule CD146 and two distinct transcription factors: HIF-1a and NF-kappa B for hypoxic reprogramming of PASMCs in Pulmonary Hypertension (PH). The study suggests that the adhesion molecule CD146 "couples" two distinct transcription factors: HIF-1 α and NF-kappa B, which signal hypoxia as well as MCT driven PH. Furthermore, the authors show that the HIF-1a serves as a target of NF-kappa B transcriptional activity in hypoxic PASMCs.

Mechanistic evidence revealed that CD146 and HIF-1a support each other. Interestingly, the activation of HIF-1a, that physiologically enables PASMC to adapt to a more synthetic phenotype, occurs via an NF-kappa B-driven and CD146-dependent pathway. The authors newly demonstrate an increase of HIF-1 α mRNA due to the NF-kappa B action - which serves as transcriptional factor maintaining HIF-1 α mRNA expression in hypoxia in the context of PH. Disruption of CD146-HIF-1 α cross-regulation by genetic ablation of CD146 in SMCs or by blockage of CD146 with monoclonal antibodies diminishes pulmonary vascular remodeling in chronically hypoxic mice, and antibody treatment prevents from further development of hypoxia and MCT-induced PH.

Major comments:

As the authors intend to transfer their findings to human PAH, investigation in biobank lung material from PAH patients and healthy controls is needed to substantiate their findings. A first step was already taken, investigating human PASMC, but expression, location and regulation of at least CD146 should be shown in human lung material.

The authors are asked to be more precise when using the terms PAH and PH. PAH (pulmonary arterial hypertension) is clearly defined in the classification of the WHO meetings (lastly updated in Nice 2018). In contrast, the hypoxia mouse model clearly represents group 3 of the classification, which is NOT PAH. Thus, the term PH should be used when displaying and discussing data from this model. Also, I am not sure whether the MCT model should be termed PAH although it reflects the PAH pathobiology in human PAH much better than hypoxia.

The authors should be very cautious discussing reversal of PH in the animal models of PH when using the antibody treatment. From the data shown for nearly all parameters only the progression of the disease is stopped, but there is no reversal towards the values of the healthy control. This is true for

hypoxia as well as MCT. As an example, in figure 6 d,f RVSP and RV hypertrophy are kept at the level of the two weeks control group and do not reach the level of the normoxic animals - whereas the isotype control treatment shows further progression of the disease. This is also true for the MCT rat model.

This fact should be clearly acknowledged in the discussion and a statement about possible side effects of the antibody treatment as well as feasibility of such a treatment for humans should be included.

To my knowledge, the discovery that hypoxia is able to activate NF-kappa B in both human and mouse PASMCs is novel and interesting, supporting an observation that NF-kappa B is activated in PASMC in patients with end-stage idiopathic PAH (Price LC et al., PLoS One. 2013). Is NF-kappa B (and also CD146) known to be induced also in the presence of various PH related pro-hypertensive growth factors and cytokines (like PDGF-BB, IL6, TNF α) or they are exclusively regulated by hypoxia in SMCs?

Does CD146 also mediate the NF-kappa B activation in response to the abovementioned growth factor/s in SMCs? Clarification of this point may further substantiate the role of the CD146-NF-kappa B axis as a driver of HIF-1 α in the pulmonary vascular remodeling process, independent of the established role of NF-kappa B in the inflammatory context of PH.

CD146 is an adhesion molecule known to be prominently expressed in T cells and endothelial cells, facilitating cell-cell interactions at endothelial cell junctions. It would thus seem reasonable to speculate that CD146 mAb will act at least as prominently via endothelial cells, lymphocytes or via smooth muscle cells in PH, though the strongest expression has been noted in SMCs. Furthermore, a large body of literature has stressed the role of the immune system in PH and lung vascular remodeling. Involvement of CD146 in the regulation of an aberrant immune response, by means of T cell activation, has also been reported. Does the CD146 antibody have an effect on accumulation of immune cells in the vascular media and adventitia in PH?

Do the authors also observe a decrease in NF-kappa B activation, by means of pp65, in the PASMCs of lungs from CD146-SMC/KO mice after hypoxic exposure in comparison to CD146 WT controls?

Figures 5,6. Please include statistics for the comparison between healthy (normoxic) and PH animals (hypoxia). Similarly, statistics comparing the MCT and healthy control groups in Figure 7 should be given.

When displaying RV/(LV+S) you should give information about possible changes in the LV. – And, if that is altered, relate the RV mass to e.g. body weight and tibia lengths.

Minor comments:

It should be mentioned and specified in the text that the activation of HIF-1 α by NF-kappa B has been already observed by Patrick van Uden et al., in Biochem J. 2008. The authors need to specify that they have confirmed the presence of the previously identified NF-kappa B-binding site located at -197/188 bp from the initiation site of the HIF-1 α sequence also in PSMCs. Unfortunately, the authors do not mention this clearly and give the reader the impression that this finding is novel.

There is no representation of VSMC/PCNA and VSMC/TUNEL staining on Figure 6, although the Figure Legend of Figure 6 clearly describes the staining. The authors should include the representative pictures of the mentioned staining into Figure 6.

All the Western Blots in Figures 2, 3 and 4 n=2 are very small. Please provide replicates of at least three independent experiments and a quantitative densitometric analysis.

In general, the manuscript should be proofread thoroughly to correct typos/mistakes. For example, Lines 279-282 "These results indicate that activation of CD146-HIF-1 α axis in SMC on hypoxic PH results in both anti-proliferative and pro-apoptotic effects on PSMCs". The authors perhaps mean to say "deactivation of CD146-HIF α axis in SMC on hypoxic PH results in both anti-proliferative and pro-apoptotic effects on PSMCs.

Line 449, page 16 / remodeling instead of remodling

Line 724, Page 26, One Way Anova instead of One Way Naova.

Line 289, Page 11, alveoli instead of veoli.

Line 20: "right heart failure" instead of "heart failure"

Line 87 "PSMC" instead of "SMC"

Reviewer #2 (Remarks to the Author):

In this article, authors describe a novel mechanistic insights into a CD146-HIF1 α loop during hypoxia that drives vascular reprogramming and PAH. They extend their results to therapy by showing that targeting this axis with an anti-CD146 antibody reduces vascular remodelling and PAH in two different animal models.

The study is well conducted and represents a lot of work. The methods are well described and the statistical analysis are relevant.

The findings are of interest in particular in this pathology in which the therapeutic treatments are missing.

However, before publication in a journal of the level of Nature Communications, several additional experiments and modifications should be included:

Major points:

- An essential point is that there is no clinical relevance in this study whereas authors claim that they want to use their antibody for therapeutic purposes. Authors should clearly indicate in the discussion section to what type of PAH patients this therapy could be useful and data obtained from human should be included in the manuscript to confirm data obtained on animals.
- Authors only show and analyze the role of CD146 and HIF1a on SMC. But CD146 is also expressed on other cells of the vessels as endothelial cells and pericytes. And these two cell types are also highly involved in vascular remodeling. Authors should include data on the regulation of CD146 and HIF1a in these two different cell types. They should also use CD146 EC-Ko mice to clearly identify the role of CD146 expressed in EC in the process.
- A time-course of the effect of hypoxia on the generation of HIF1a and CD146 should be given in the three cell types.
- Authors show a nice effect of the anti-CD146 antibodies to prevent vascular remodeling and PAH. They should also analyze the potential interest of their antibody in reverting the process by treating the animals after the establishment of the pathology.

Minor:

- Authors show no expression of CD146 on human PAEC. This is very strange and should be verified with other anti-CD146 antibodies.
- Histological and immunofluorescence pictures should be enlarged to better evaluate the effects.

Dr Blot-Chabaud

C2VN. Inserm 1263 Aix-Marseille University

Marseille- France

Point-by-point response

We would like to sincerely thank both reviewers for their careful and critical review of our manuscript. We are also grateful to the reviewers for their encouraging comments and insightful suggestions that allowed us to substantially improve our manuscript. In light of these comments, we have conducted a significant number of additional experiments and introduced textual changes to the manuscript. All the figure numbers are those of the revised manuscript, except Figures R1-R11 that correspond to the figures found only in this rebuttal. Revisions and changes to the manuscript are clearly highlighted in red. Following is our point-by-point response.

Reviewer #1 (Remarks to the Author):

The present study uncovered a previously unrecognized cross-regulation between the adhesion molecule CD146 and two distinct transcription factors: HIF-1 α and NF-kappa B for hypoxic reprogramming of PSMCs in Pulmonary Hypertension (PH). The study suggests that the adhesion molecule CD146 "couples" two distinct transcription factors: HIF-1 α and NF-kappa B, which signal hypoxia as well as MCT driven PH. Furthermore, the authors show that the HIF-1 α serves as a target of NF-kappa B transcriptional activity in hypoxic PSMCs. Mechanistic evidence revealed that CD146 and HIF-1 α support each other. Interestingly, the activation of HIF-1 α , that physiologically enables PSMC to adapt to a more synthetic phenotype, occurs via an NF-kappa B-driven and CD146-dependent pathway. The authors newly demonstrate an increase of HIF-1 α mRNA due to the NF-kappa B action - which serves as transcriptional factor maintaining HIF-1 α mRNA expression in hypoxia in the context of PH. Disruption of CD146-HIF-1 α cross-regulation by genetic ablation of CD146 in SMCs or by blockage of CD146 with monoclonal antibodies diminishes pulmonary vascular remodeling in chronically hypoxic mice, and antibody treatment prevents from further development of hypoxia and MCT-induced PH.

Response: We sincerely thank the reviewer for the encouraging and insightful comments.

Major comments:

Comment 1: *As the authors intend to transfer their findings to human PAH,*

investigation in biobank lung material from PAH patients and healthy controls is needed to substantiate their findings. A first step was already taken, investigating human PASMC, but expression, location and regulation of at least CD146 should be shown in human lung material.

Response: We thank the Reviewer for this suggestion. We fully agree with the reviewer, and performed additional experiments using human lung material and the new data have been shown in revised Figure 1i-l.

To explore and compare the differential expression of CD146 in pulmonary arteries from patients with idiopathic pulmonary arterial hypertension (IPAH) and healthy controls, we performed immunofluorescent staining of lung sections, and the representative images are shown in our revised Figures 1i and k, with the quantification of CD146 expression and localization shown in revised Figures 1j and l. We found that CD146 expression of human PAH lungs is upregulated, and mainly occurs in the medial layer of PASMCs in the remodeled small pulmonary arteries as identified by co-staining with α SMA. However, CD146 expression in the neointima was almost undetectable (revised Figure 1i-l). This expression pattern of CD146 in human PAH lungs is consistent with the pattern observed in mouse and rat PH lungs, suggesting that the upregulation of CD146 in PASMCs is a common feature of pulmonary hypertension.

***Comment 2:** The authors are asked to be more precise when using the terms PAH and PH. PAH (pulmonary arterial hypertension) is clearly defined in the classification of the WHO meetings (lastly updated in Nice 2018). In contrast, the hypoxia mouse model clearly represents group 3 of the classification, which is NOT PAH. Thus, the term PH should be used when displaying and discussing data from this model. Also, I am not sure whether the MCT model should be termed PAH although it reflects the PAH pathobiology in human PAH much better than hypoxia.*

Response: We thank the Reviewer for pointing out this issue. We carefully studied the World symposium on pulmonary hypertension (WSPH) in Nice 2018 (*Eur Respir J* 2019; 53:1801913), and the state-of-the-art reviews (*BMJ* 2018, 360:j5492; *Eur Respir J.* 2019, 24;53(1)) on classification of pulmonary hypertension. We fully agree with the reviewer, and replaced PAH with PH when describing data from hypoxia mouse model throughout the manuscript.

Regarding the MCT model, so far no clearly defined classification has been described, although this model was assigned to group 1 PH in 2011 (*Int J Clin Pract Suppl.* 2011, (172):15-34). Even when we searched the literature, we found that both PAH (*Nat Commun.* 2017, 8: 14079; *Nat Med.* 2015, 21(7):777-85; *Nat Med.* 2014, 20(11):1289-300; *Cell Metab.* 2014, 20(5):827-839) and PH (*Nature.* 2015, 524(7565):356-60; *Nat Commun.* 2018, 9: 3850; *Nat Med.* 2009, 15(11):1289-97; *Nat Med.* 2013, 19(1):74-82) were used for describing MCT models. We agree with the reviewer that, although MCT model better reflects and recapitulates several features of human PAH, the term PAH should be used cautiously for the MCT model, due to its distinct natural history and etiology compared to human PAH. Therefore, we opt to use PH when describing the MCT model in this manuscript.

Comment 3: *The authors should be very cautious discussing reversal of PH in the animal models of PH when using the antibody treatment. From the data shown for nearly all parameters only the progression of the disease is stopped, but there is no reversal towards the values of the healthy control. This is true for hypoxia as well as MCT. As an example, in figure 6 d,f RVSP and RV hypertrophy are kept at the level of the two weeks control group and do not reach the level of the normoxic animals - whereas the isotype control treatment shows further progression of the disease. This is also true for the MCT rat model. This fact should be clearly acknowledged in the discussion and a statement about possible side effects of the antibody treatment as well as feasibility of such a treatment for humans should be included.*

Response: Thank you for this suggestion. We fully agree with the reviewer that antibody treatment in both models impedes disease progression towards a more severe phenotype, without completely reversal of the disease towards a healthy state. We have inserted additional text into the *Discussion* to clearly acknowledge this fact (page 14).

With regard to the possible side effects and feasibility of this antibody treatment for human PAH, we have inserted additional text and references into *Discussion* to address this important issue (pages 15 and 16).

The potential side effects of an antibody-based therapy should be dependent on the distribution of the CD146 protein in normal tissues. Previous studies, including ours, showed that CD146 is expressed in a small subset of peripheral T lymphocytes (*Sci*

Rep. 2013, 3:1687; *Am J Pathol.* 2014, 184(5):1604-16; *Autoimmun Rev.* 2015, 14(5):415-22). Moreover, CD146 is also present in the vascular wall with spatially dynamic localization, including microvascular endothelial cells without perivascular cell coverage, and perivascular cells (SMCs and pericytes) in normal tissues (*Proc Natl Acad Sci U S A.* 2017, 114(36):E7622-E7631; *Blood.* 2003, 1;102(1):184-91). Because CD146 has been implicated in vascular development, cell differentiation, migration, signal transduction, and T-cell activation (*Arterioscler Thromb Vasc Biol.* 2019, 18:ATVBAHA119312653; *Cancer Lett.* 2013, 330(2):150-62), one might expect side effects caused by such anti-CD146 therapy. We have previously reported that anti-CD146 AA98 does neither affect the proliferation, differentiation and activation of lymphocyte nor other CD146⁺ immune cells from both human and mice (*Sci Rep.* 2013, 3:1687; *Am J Pathol.* 2014, 184(5):1604-16), suggesting that AA98 might not elicit adverse effects by sparing the host's immune response. Regarding vascular wall, we previously screened 48 normal tissues and observed that AA98 did not recognize, or only slightly recognized, blood vessels in normal tissues with very low frequency (18.8%). In the few cases where AA98 recognized vessels, only perivascular cells were stained (*Blood.* 2003, 1;102(1):184-91). Although AA98 has a very low propensity to recognize normal blood vessels, we still cannot exclude possible side effects of the antibody treatment by targeting vascular wall. This safety issue needs to be addressed in future studies.

Apart from potential side effects, effectiveness of the antibody treatment must be verified before translating this therapy for human PAH. In this study, we provide evidence that CD146 is elevated in small PAs from human PAH lungs; anti-CD146 impeded the switch of human PSMCs towards a more synthetic phenotype. These results provide a rationale for the potential application of this antibody-based therapy for human PAH. It is intriguing to postulate that anti-CD146 antibody might be efficacious across a spectrum of pulmonary hypertension etiologies, e.g., IPAH and PH associated with hypoxemia, since smooth muscle expansion represents a common pathological hallmark of the disease. Further efforts to determine the effectiveness of anti-CD146 treatment, as well as understanding potential side effects, should lead to new therapeutics for human PAH.

Comment 4: *To my knowledge, the discovery that hypoxia is able to activate NF-kappa B in both human and mouse PSMCs is novel and interesting, supporting*

an observation that NF-kappa B is activated in PASMC in patients with end-stage idiopathic PAH (Price LC et al., PLoS One. 2013). Is NF-kappa B (and also CD146) known to be induced also in the presence of various PH related pro-hypertensive growth factors and cytokines (like PDGF-BB, IL6, TNF α) or they are exclusively regulated by hypoxia in SMCs? Does CD146 also mediate the NF-kappa B activation in response to the abovementioned growth factor/s in SMCs? Clarification of this point may further substantiate the role of the CD146-NF-kappa B axis as a driver of HIF-1 α in the pulmonary vascular remodeling process, independent of the established role of NF-kappa B in the inflammatory context of PH.

Response: We sincerely thank the reviewer for the constructive comment. Following the reviewer's suggestion, we stimulated mouse PASMCs using these pro-hypertensive factors as well as hypoxia, and measured how NF- κ B and CD146 respond and how CD146 deficiency modulates NF- κ B activation. As a result, we observed that CD146 and NF- κ B showed distinct responsive patterns under these stimuli. Stimulation with PDGF-BB, TNF- α and hypoxia, but not IL-6, led to an increase in NF- κ B activation in PASMCs isolated from CD146^{WT} mice. However, distinct from hypoxia, stimulation with pro-hypertensive factors showed no effect on CD146 expression. In addition, CD146 deficiency impaired NF- κ B activation induced by PDGF-BB and hypoxia, but not TNF- α (Figure R1). Together, these results clearly indicate that TNF- α -dependent NF- κ B activation is conserved and is different from that of the PDGF-BB- or hypoxia-dependent NF- κ B activation pathway. This finding is in agreement with our previous reports showing that CD146 functions as a co-receptor for PDGFR β and promotes PDGF-PDGFR β signaling (*Proc Natl Acad Sci U S A.* 2017, 114(36):E7622-E7631; *Protein Cell.* 2018, 9(8):743-747), and CD146 is not involved in TNF- α -induced NF- κ B activation (*Blood.* 2012, 120(11):2330-9). Therefore, the differential expression pattern of NF- κ B and CD146 in response to distinct stimuli suggests an exclusively regulated pattern of CD146-NF- κ B axis by hypoxia in PASMCs. We have now included these results into the Figure R1.

Comment 5: *CD146 is an adhesion molecule known to be prominently expressed in T cells and endothelial cells, facilitating cell-cell interactions at endothelial cell junctions. It would thus seem reasonable to speculate that CD146 mAb will act at least as prominently via endothelial cells, lymphocytes or via smooth muscle cells in*

PH, though the strongest expression has been noted in SMCs. Furthermore, a large body of literature has stressed the role of the immune system in PH and lung vascular remodeling. Involvement of CD146 in the regulation of an aberrant immune response, by means of T cell activation, has also been reported. Does the CD146 antibody have an effect on accumulation of immune cells in the vascular media and adventitia in PH?

Response: We thank the reviewer for this suggestion. The effect of the CD146 antibody on accumulation of immune cells in PAs in PH is indeed an interesting point. As suggested, we performed immunofluorescent staining of lung samples from hypoxic mice or MCT rats treated with anti-CD146 antibody or isotype control. We measured the number of lymphocytes and macrophages infiltrating pulmonary arteries, which have been frequently observed and implicated in the development of both experimental PH and clinical PAH (*Circ Res.* 2014, 20;115(1):165-75; *PLoS One.* 2013, 4;8(10):e75415). As shown in Figure R2, anti-CD146 AA98 treatment showed no significant effects on the accumulation of CD3⁺ lymphocytes or Mac-3⁺ macrophages in hypoxia- or MCT-induced inflammatory cell infiltration around the small pulmonary arteries (Figure R2). These results suggest that the anti-remodeling effects of the anti-CD146 antibody might be mainly attributed to its inhibitory role on CD146-HIF-1 α -driven PASMC expansion, leaving perivascular inflammatory response intact. Our previous studies showed that, in various inflammatory conditions, such as neuroinflammation and colitis, the anti-CD146 AA98 inhibited immune cell infiltration by targeting CD146 on microvascular endothelial cells but not immune cells (*Sci Rep.* 2013, 3:1687; *Am J Pathol.* 2014, 184(5):1604-16). Therefore, the inability of this antibody in dampening perivascular inflammation of PAs might be due to the absence of CD146 expression in pulmonary arterial endothelial cells in PH (Figure 1i-l).

Comment 6: *Do the authors also observe a decrease in NF-kappa B activation, by means of pp65, in the PASMCs of lungs from CD146-SMC/KO mice after hypoxic exposure in comparison to CD146 WT controls?*

Response: As suggested by the reviewer, we performed additional experiments, and the new data has been added in revised Figure 5b, showing the decrease in NF-kappa B activation (by means of p-p65) in the PASMCs of lungs from CD146^{SMC-KO} mice after hypoxic exposure.

Comment 7: *Figures 5,6. Please include statistics for the comparison between healthy (normoxic) and PH animals (hypoxia). Similarly, statistics comparing the MCT and healthy control groups in Figure 7 should be given.*

Response: We have now included the statistics for the comparison between healthy and PH animals in the revised Figures 5-7.

Comment 8: *When displaying $RV/(LV+S)$ you should give information about possible changes in the LV. – And, if that is altered, relate the RV mass to e.g. body weight and tibia lengths.*

Response: We agree with the reviewer that the Fulton index $RV/(LV+S)$ would be affected by fluctuations in the LV. As suggested, we analyzed the LV mass in both hypoxia and MCT models, and found that there were no significant changes in LV mass in hypoxic mouse models. However, the LV mass was slightly and significantly reduced after MCT induction in rats (Figure R3). These results are consistent with those shown in other studies (*Am J Physiol Heart Circ Physiol.* 2006, 291(2):H507-16; *Physiol Rep.* 2013, 15;1(7):e00184; *J Am Coll Cardiol.* 2011, 22;57(8):921-8; *Cardiovasc Res.* 2017, 113(1):15-29; *Br J Pharmacol.* 2019, 176(9):1206-1221). In conditions in which LV mass changes, RV hypertrophy can be more accurately quantified by relating RV weight to tibia length (TL) or body weight (BW) (*Am J Physiol.* 1982, 243(6):H941-7; *J Am Coll Cardiol.* 2011, 22;57(8):921-8). Therefore, we determined RV hypertrophy by calculating RV/TL and RV/BW in the MCT model. As shown in supplementary Figure 11, antibody treatment inhibited RV hypertrophy in MCT model as evidenced by RV/TL and RV/BW .

Minor comments:

Comment 1: *It should be mentioned and specified in the text that the activation of HIF-1 α by NF-kappa B has been already observed by Patrick van Uden et al., in Biochem J. 2008. The authors need to specify that they have confirmed the presence of the previously identified NF-kappa B-binding site located at –197/188 bp from the initiation site of the HIF-1 α sequence also in PSMCs. Unfortunately, the authors do not mention this clearly and give the reader the impression that this finding is novel.*

Response: We agree with the reviewer and apologize for not clarifying this in the first version of the manuscript. As suggested, we cited this reference and included this point in the text (page 8).

Comment 2: *There is no representation of VSMC/PCNA and VSMC/TUNEL staining on Figure 6, although the Figure Legend of Figure 6 clearly describes the staining. The authors should include the representative pictures of the mentioned staining into Figure 6.*

Response: As suggested, we have added the representative pictures of VSMC/PCNA and VSMC/TUNEL staining into the revised Figure 6.

Comment 3: *All the Western Blots in Figures 2, 3 and 4 n=2 are very small. Please provide replicates of at least three independent experiments and a quantitative densitometric analysis.*

Response: In light of this comment, we performed additional experiments for the Western Blots, and the representative blots has been shown in Figures 2-4, and the quantitative densitometric analysis in Figures 2, 3 and Supplementary Fig. 5 and 6 to confirm the reproducibility (n = 3).

Comment 4: *In general, the manuscript should be proofread thoroughly to correct typos/mistakes. For example, Lines 279-282 “These results indicate that activation of CD146-HIF-1 α axis in SMC on hypoxic PH results in both anti-proliferative and pro-apoptotic effects on PASMCs”. The authors perhaps mean to say “deactivation of CD146-HIF α axis in SMC on hypoxic PH results in both anti-proliferative and pro-apoptotic effects on PASMCs.*

Line 449, page 16 / remodeling instead of remodling

Line 724, Page 26, One Way Anova instead of One Way Naova.

Line 289, Page 11, alveoli instead of veoli.

Line 20: “right heart failure” instead of “heart failure”

Line 87 “PASMC” instead of “SMC”

Response: As suggested, we had the entire manuscript professionally edited and polished by an English-language editing service. All the spelling and grammatical errors, including above mentioned typos/mistakes, have been fixed.

Reviewer #2 (Remarks to the Author):

In this article, authors describe a novel mechanistic insights into a CD146-HIF1 α loop during hypoxia that drives vascular reprogramming and PAH. They extend their results to therapy by showing that targeting this axis with an anti-CD146 antibody reduces vascular remodelling and PAH in two different animal models.

The study is well conducted and represents a lot of work. The methods are well described and the statistical analysis are relevant.

The findings are of interest in particular in this pathology in which the therapeutic treatments are missing.

However, before publication in a journal of the level of Nature Communications, several additional experiments and modifications should be included:

Response: Reviewer #2 recognizes the novelty and the potential clinical impact of this study. We sincerely thank the reviewer for this very encouraging comment.

Major points:

Comment 1: *An essential point is that there is no clinical relevance in this study whereas authors claim that they want to use their antibody for therapeutic purposes. Authors should clearly indicate in the discussion section to what type of PAH patients this therapy could be useful and data obtained from human should be included in the manuscript to confirm data obtained on animals.*

Response: We thank the reviewer for this suggestion. Following the reviewer's suggestion, we inserted additional text into the *Discussion* to address the type of PAH this therapy could be applicable (page 16). We also performed additional experiments using human samples, and the new data have now been included in the revised Figure 1i-l.

To explore and compare the differential expression of CD146 in pulmonary arteries from patients with idiopathic pulmonary arterial hypertension (IPAH) and healthy

controls, we performed immunofluorescent staining of lung sections. Representative images are shown in revised Figures 1i and k, with quantification of CD146 expression and localization shown in the revised Figures 1j and l. We found that CD146 expression of human PAH lungs was upregulated, mainly occurring in the medial layer of PASMCs in the remodeled small pulmonary arteries as identified by co-staining with α SMA. However, CD146 expression in the neointima was almost undetectable (revised Figure 1i-l). This expression pattern of CD146 in human PAH lungs is consistent with the pattern observed in mouse and rat PH lungs, suggesting that the upregulation of CD146 in PASMCs is a common feature of pulmonary hypertension.

***Comment 2:** Authors only show and analyze the role of CD146 and HIF1 α on SMC. But CD146 is also expressed on other cells of the vessels as endothelial cells and pericytes. And these two cell types are also highly involved in vascular remodeling. Authors should include data on the regulation of CD146 and HIF1 α in these two different cell types. They should also use CD146 EC-Ko mice to clearly identify the role of CD146 expressed in EC in the process.*

Response: We thank the reviewer for the suggestions. We performed additional experiments to determine the regulation of CD146 and HIF-1 α in pulmonary endothelial cells and pericytes. We also employed a CD146^{EC-KO} mouse model to determine the role of endothelial CD146 in the development of hypoxia-induced PH. The new data have now been added to the Figures R4-R8.

Endothelial cells lining pulmonary vascular tree are differentially specialized and display remarkable heterogeneity in molecular signature (*Circ Res.* 2007, 2;100(2):174-90.; *Nat Rev Mol Cell Biol.* 2017, 18(8):477-494). In order to determine the regulation of CD146 and HIF-1 α in pulmonary endothelial cells, as well as pericytes, we first investigated the expression pattern of CD146 in the two types of cells in mouse lung tissues. As for pulmonary arteries, CD146 expression was almost undetectable in PAECs (Figure 1i-l). We next determined the expression pattern of CD146 in endothelial cells and pericytes of pulmonary microvasculature by performing 3D-reconstruction post-confocal imaging of lung sections after immunofluorescent staining of CD146, endothelial marker CD31, and perivascular markers α SMA and PDGFR β . Our analysis showed that in pulmonary microvasculature, CD146 was mainly expressed in pericytes, with little expression

detected in pulmonary microvascular endothelial cells (PMVECs) covered with pericytes. In contrast, PMVECs lacking pericyte coverage still expressed CD146 (Figure R4). This expression pattern of CD146 is consistent with that observed in cerebral microvasculature (*Proc Natl Acad Sci U S A.* 2017, 114(36):E7622-E7631). Therefore, we isolated PMVECs and pericytes from CD146^{WT} and CD146^{KO} mice as previously described (*Am J Physiol.* 1994, 267:433-41; *Sci Pages Pulmonol.* 2017, 1: 7-18; *Proc Natl Acad Sci U S A.* 2017, 114(36):E7622-E7631), and investigated the regulation of CD146 and HIF-1 α under hypoxia. We found that treatment with hypoxia induces HIF-1 α expression in both PMVECs and pericytes isolated from CD146^{WT} mice. Hypoxia induced upregulation of CD146 in PMVECs, not however in pericytes. In addition, CD146 deficiency in both cell types showed no effect on hypoxia-induced HIF-1 α expression (Figure R5). Together, these results suggested that the cross-regulation between CD146 and HIF-1 α might be confined within PASMCs but not other types of vascular cells.

To further determine whether endothelial CD146 plays a role in pulmonary vascular remodeling as requested by the reviewer, we studied the development of hypoxic PH using CD146^{EC-KO} mice. We previously reported that these knockout mice were viable and fertile, and exhibit normal development of peripheral blood vessels (*Protein Cell.* 2014, 5(6):445-56; *Am J Pathol.* 2014, 184(5):1604-16). In lung tissues, endothelial CD146 deficiency showed no effects on pulmonary vessel development and integrity (Figure R6). Based on these results, we next studied the development of hypoxic PH using CD146^{EC-KO} mice (Figure R7a). After 4-week hypoxia induction (10% O₂), we detected no changes in lung hemodynamics, RV hypertrophy and small pulmonary arterial remodeling between the two groups (Figure R7b-m). The lack of effects of endothelial CD146 on the development of pulmonary arterial remodeling might be attributable to the absence of CD146 expression in PAEC (Figure 1i-l).

Although there was no significant difference in PH development following chronic hypoxia, we noted a more impressive phenotype upon an additional one month recovery (reoxygenation) following hypoxia. We found that CD146^{EC-KO} mice demonstrated persistently elevated RVSP, TPVRI, RV hypertrophy and RV dysfunction after 1 month in recovery (Figure R7b-j). We propose that the persistent elevation of RVSP and RV failure in CD146^{EC-KO} mice in recovery might be due to abnormalities in pulmonary microvascular structure linked to increased vascular resistance and greater cardiac afterload. Indeed, consistent with the hemodynamic data, WT mice demonstrated a significant reduction in muscularization of distal

pulmonary arteries upon return to normoxia, whereas no change was noted in CD146^{EC-KO} mice (Figure R7k-m). Further assessment demonstrated a significant reduction in number of distal vessels in CD146^{EC-KO} mice after recovery (Figure R7n), suggesting that the CD146^{EC-KO} mice have an impaired capacity to regenerate pulmonary microvasculature. In fact, PMVECs isolated from CD146^{EC-KO} mice after recovery versus WT mice showed a pro-apoptotic and anti-proliferative phenotype (Figure R8a-c). Migration and tube formation of PMVECs from CD146^{EC-KO} mice was also impaired (Figure R8d,e). These features are consistent with compromised regeneration of microvasculature in hypoxia-reoxygenation.

Collectively, these results imply that CD146 expression, probably in PMVECs, might contribute to the progressive vessel remodeling in PAH through preservation and proper maintenance of the pulmonary microcirculation. Nevertheless, we think that addressing the precise role of endothelial CD146 in PH lies outside the scope of this manuscript; such studies should be included in future investigations.

***Comment 3:** A time-course of the effect of hypoxia on the generation of HIF1 α and CD146 should be given in the three cell types.*

Response: We thank the reviewer for this constructive comment. As suggested, we treated PSMCs, PMVECs and pericytes with hypoxia for the indicated time points, and then measured the expression levels of HIF-1 α and CD146 by immunoblotting. The three cell types exhibited distinct responsive patterns to hypoxia, both for HIF-1 α and CD146 expression. We found that the responsiveness of HIF-1 α and CD146 to hypoxia was most prominent in PSMCs, and less robust in PMVECs. Although hypoxia induced HIF-1 α expression at a latter time point in pulmonary pericytes, there was no apparent elevation of CD146 expression (Figure R9). This result, together with the result showing in Figure R5, suggested that the cross-regulation between CD146 and HIF-1 α might be specific to PSMCs.

***Comment 4:** Authors show a nice effect of the anti-CD146 antibodies to prevent vascular remodeling and PAH. They should also analyze the potential interest of their antibody in reverting the process by treating the animals after the establishment of the pathology.*

Response: We thank the reviewer for this comment. Analyzing the potential effect of

the anti-CD146 antibody in the treatment of established PAH has already been performed in two PH models in this study, and the therapeutic data has been displayed in Figure 6 (hypoxia model) and Figure 7 (MCT model). The antibody treatment was commenced 2 weeks after hypoxia induction in mice and 3 weeks post MCT exposure in rats, a time point at which PH pathology was established (Figure 6 and 7). We found that anti-CD146 treatment after PH disease onset significantly increased lung hemodynamics, reduced RV hypertrophy, improved right-heart function, and suppressed remodeling of the pulmonary arteries. It should be noted that antibody therapy in both models impedes disease progression towards a more severe phenotype, without completely reversal of the disease towards a healthy state. To make this issue clearer, we have now inserted additional text into the *Discussion* of the revised version of our manuscript (page 14).

Minor:

Comment 1: *Authors show no expression of CD146 on human PAEC. This is very strange and should be verified with other anti-CD146 antibodies.*

Response: We thank the reviewer for this comment. As suggested, we used additional commercially available anti-CD146 antibody to verify the absence of CD146 on human PAEC, and the data are now included in the Figures R10-11 and in the paper as Figure 1i-l.

To accurately explore and distinguish the expression pattern of CD146 in PAEC and PASMC, we performed 3D-reconstruction post-confocal imaging of lung sections from human healthy subjects after immunofluorescent staining of CD146 (ab10816, abcam), endothelial marker CD31 (ab56299, abcam), and arterial markers α SMA (ab21027, abcam) and SMMHC (ab53219, abcam). The reconstructed 3D images enabled us to capture and analyze the expression and location of CD146 in pulmonary arteries with differential vessel diameters. As a result, we observed that CD146 expression in pulmonary arteries with differential vessel diameters (20-100 μ m) was prominently confined to the medial layer of PASMCs as identified by co-staining with α SMA and SMMHC, but undetectable in endothelial cells of pulmonary arteries with complete SMC ensheathment (Figure R10). In addition, this perivascular location of CD146 in human pulmonary arteries was also observed in human PH lungs and further confirmed in mouse and rat lungs (Figure 1i-l) as stained by different

anti-CD146 antibodies (ab75769/abcam for human and rat CD146, ME-9F1/Biolegend for mouse CD146). This expression pattern of CD146 in pulmonary arteries is consistent with that observed in arteries of brain, liver, pancreas, placenta, and adipose tissues as reported in our previous studies (*Proc Natl Acad Sci U S A.* 2017, 114(36):E7622-E7631; *Blood.* 2003, 1;102(1):184-91) as well as others (*Nature.* 2018, 563(7731):347-353; *Cell Stem Cell.* 2008, 3(3):301-13). The absence of CD146 in endothelial cells lining arteries might be due to perivascular cell-mediated CD146 downregulation through TGF- β 1 during vascular maturation (*Proc Natl Acad Sci U S A.* 2017, 114(36):E7622-E7631).

In this study, we observed that CD146 expression in the primary isolated PAECs (Passage 0) was undetectable by immunoblotting (Figure 1h), consistent with that observed in lung tissues (Figure 1i-l). However, we further noted the appearance of CD146 expression along cell passages (beginning at passage 2) (Figure R11), which might be the result from the lack of perivascular cell-mediated suppression of CD146 expression in *in vitro* culture conditions.

Comment 2: *Histological and immunofluorescence pictures should be enlarged to better evaluate the effects.*

Response: As suggested by the reviewer, we enlarged all histological and immunofluorescence images.

Figures only for Response

Fig. R1. CD146 and NF- κ B showed distinct responsive patterns under hypertensive factors stimulation and hypoxia. Primary murine PASCs (CD146^{+/+} and CD146^{-/-}) were stimulated by hypertensive factors or cultured under hypoxia. The expression of CD146 and NF- κ B p-p65 was measured by immunoblotting. Below, quantification of CD146 and p-p65 expression. In all statistical plots, the results are expressed as mean \pm s.e.m. ** $P < 0.01$, *** $P < 0.001$. n.s., not significant; by two-tailed Student's t-test. All WB represent data from three independent repeats.

Fig. R2. Anti-CD146 AA98 treatment showed no effects on the accumulation of CD3⁺ lymphocytes or Mac-3⁺ macrophages in hypoxia- or MCT-induced inflammatory cell infiltration around the small pulmonary arteries. Immunostaining of CD3 and Mac-3 of lungs from hypoxia- or MCT-treated animals. White arrows, positive staining. α SMA, red. DAPI, blue. Scale bar, 30 μ m.

Fig. R3. LV mass in CD146^{WT} and CD146^{SMC-KO} mice treated with normoxia or hypoxia (a), or hypoxia-induced CD146^{WT} mice preventively (b) or therapeutically (c) treated with anti-CD146 antibody or isotype control, or MCT-induced rats therapeutically (d) treated with anti-CD146 antibody or isotype control (10 animals per group). In all statistical plots, the results are expressed as mean \pm s.e.m. n.s., not significant; by one-way ANOVA with Bonferroni *post hoc* analysis.

Fig. R4. Expression of CD146 in pulmonary microvasculature. Lung sections (20 μm thickness) were stained for CD146 (green), CD31 (EC marker, red), perivascular markers αSMA (blue) and PDGFR β (purple). Shown are three-dimensional reconstructions of confocal image z-stacks of pulmonary microvasculature. The expression of CD146 in pulmonary microvasculature was mainly in pericytes but not in pulmonary microvascular endothelial cells (PMVECs) covered with pericytes (arrowheads), whereas PMVECs without pericyte coverage still expressed CD146 (arrows). Scale bars represent 10 μm . Two representative pulmonary microvessels were shown.

Fig. R5. The cross-regulation between CD146 and HIF-1 α might not be conserved in PMVECs and pericytes. PMVECs and pulmonary pericytes isolated from CD146^{WT} or CD146^{KO} mice were treated under normoxia or hypoxia for 24h and the expression of CD146 and HIF-1 α was measured by Western Blot. Below, quantification of CD146 and HIF-1 α expression. In all statistical plots, the results are expressed as mean \pm s.e.m. *** $P < 0.001$. n.s., not significant; by two-tailed Student's t-test. All WB represent data from three independent repeats.

Fig. R6. Neither vascular number nor integrity was altered in the lung from CD146^{EC-KO} mice. **(a)** Quantification of the ratio of vessel/ 100 alveoli in the lung of CD146^{WT} and CD146^{EC-KO} mice (n = 6 mice per group). **(b,c)** Wet/Dry Weight Ratio **(b)** and water content **(c)** of the lung from 8-10 week old CD146^{WT} and CD146^{EC-KO} mice (n = 6 mice per group). **(d-f)** CD146^{WT} and CD146^{EC-KO} mice (8-10 week) were given an i.p. injection or i.v. injection of Evans blue dye or FITC-dextran-4 kD and FITC-dextran-70 kD dye respectively, and the absorption of the dyes extracted from the lung was measured by a microplate spectrophotometer (n = 6 mice per group). In all statistical plots, the results are expressed as mean ± s.e.m. n.s., not significant. Two-tailed Student's t-test.

Fig. R7. CD146^{EC-KO} mice show unresolved PH in hypoxia-reoxygenation. **(a)** Schematic of hypoxia-induced PH and recovery in CD146^{EC-KO} and CD146^{WT} mice. **(b-e)** RVSP, RV/(LV + S), TPVRI and SAP in CD146^{EC-KO} and CD146^{WT} mice in normoxia, hypoxia, and recovery (n = 6 mice per group). **(f-j)** Quantification of echocardiographic (PAAT, TAPSE, RVID, CO and CI) measurements. **(k,l)** Quantification of vascular medial thickness **(k)** and PAs with > 50% luminal stenosis **(l)** (n = 4 mice per group, 3 PAs per mouse). **(m)** Proportion of non-, partially-, or fully-muscularized pulmonary arterioles (20-50 μ m in diameter) (n = 4 mice per group, 3 lung sections per mouse, 3 PAs per section). **(n)** Quantification of the number

of distal vessels per 100 alveoli (n = 6 mice per group). In all statistical plots, the results are expressed as mean \pm s.e.m. ** $P < 0.01$, *** $P < 0.001$. n.s., not significant; by one-way ANOVA with Bonferroni *post hoc* analysis (**b-j**) or two-tailed Student's t-test (**k-n**).

Fig. R8. CD146 deficiency in PMVECs results in endothelial dysfunction in response to hypoxia-reoxygenation. (a) PMVECs were isolated from CD146^{EC-KO} and CD146^{WT} mice, and apoptosis (a) and proliferation (b,c) were assessed by caspase-3/7 luminescence assay and CCK8 or BrdU incorporation assays. (d,e) Cell migration (d) and tube formation (e) were assessed by a modified Boyden chamber assay and a Matrigel tube formation assay.

Fig. R9. A time-course of the effect of hypoxia on the generation of HIF-1 α and CD146 in primary PSMCs, PMVECs and pericytes. The three types of cells isolated from CD146^{WT} mice were treated under normoxia or hypoxia for indicated time points and the expression of CD146 and HIF-1 α was measured by Western Blot. Below, quantification of CD146 and HIF-1 α expression. In all statistical plots, the results are expressed as mean \pm s.e.m. ** $P < 0.01$, *** $P < 0.001$. n.s., not significant; by two-tailed Student's t-test. All WB represent data from three independent repeats.

Fig. R10. Expression of CD146 in human pulmonary arteries. **(a,b)** Lung sections (20 μm thickness) were stained for CD146 (green), CD31 (EC marker, red), arterial markers SMMHC (blue) and αSMA (purple). Shown are three-dimensional reconstructions of confocal image z-stacks of pulmonary arteries **(a)** and the three-dimensional surface rendering of epifluorescence images **(b)**. White lines indicate cutting sites shown on the right of each panel, confirming that the expression of CD146 was exclusively observed in smooth muscle cells and not in PAECs of pulmonary arteries. Cut-open images were then created from the three-dimensional surface rendering of vessels to reveal the inner vessel wall **(b)**. **(c)** Shown are three-dimensional reconstructions of confocal image z-stacks of pulmonary arteries with differential vessel diameters. Scale bars represent 30 μm. At least 20 pulmonary arteries were analyzed.

Fig. R11. Expression of CD146 in PAECs and PSMCs in culture. CD146 expression in primary PAECs isolated from CD146^{WT} mice were was measured by Western Blot. Primary PSMCs were used as a control.

REVIEWERS' COMMENTS:

Reviewer #1 (Remarks to the Author):

All of my previous comments and concerns have been addressed.

Some of the novel data provided for the reviewers only (the detailed analyses of the expression pattern of CD146 in pulmonary arteries and pulmonary microvasculature, as well as the finding that hypoxia induced upregulation of CD146 only in PMVECs - but not in PAECs or pericytes) should be included into the manuscript.

The authors also provided an interesting novel data set of the distinct hypoxia-induced mode of cross-regulation between CD146 and HIF-1 α in PMVECs and pericytes. These data should also be included in the manuscript as a supporting information.

Taken together, I recommend including the figures to the reviewers: Fig R4, Fig R5 and Fig R10, as supporting data in the revised manuscript and discuss those novel findings in the text.

Reviewer #2 (Remarks to the Author):

The reviewers have appropriately answered to all the questions of the reviewer.

New data which have been generated are of interest.

I propose that authors integrate the new data obtained on CD146-ECKO mice in the supplementary data (figures R6 and R7) and have a short discussion about them in the paper.

M. Blot-Chabaud

C2VN, Inserm 1263, Inra 1260, Aix-Marseille University

UFR Pharmacy

Marseille, France

Point-by-point response to reviewers

We would like to sincerely thank both reviewers for their precious time and efforts in reviewing our manuscript, and highly appreciate their positive appraisal of our work. As further suggested by the reviewers, we have integrated the new data into the Results section and made additional discussion in this final version. Following is our point-by-point response.

REVIEWERS' COMMENTS:

Reviewer #1 (Remarks to the Author):

All of my previous comments and concerns have been addressed.

Some of the novel data provided for the reviewers only (the detailed analyses of the expression pattern of CD146 in pulmonary arteries and pulmonary microvasculature, as well as the finding that hypoxia induced upregulation of CD146 only in PMVECs - but not in PAECs or pericytes) should be included into the manuscript.

The authors also provided an interesting novel data set of the distinct hypoxia-induced mode of cross-regulation between CD146 and HIF-1 α in PMVECs and pericytes. These data should also be included in the manuscript as a supporting information.

Taken together, I recommend including the figures to the reviewers: Fig R4, Fig R5 and Fig R10, as supporting data in the revised manuscript and discuss those novel findings in the text.

Response: We are glad that the reviewer is satisfied with the revision. We sincerely thank the reviewer for this constructive comment. As suggested, we have incorporated the new data (Fig R4, Fig R5, Fig R9 and Fig R10) into Supplementary Information and added a very short discussion in the manuscript (pages 11).

Reviewer #2 (Remarks to the Author):

The reviewers have appropriately answered to all the questions of the reviewer.

New data which have been generated are of interest.

I propose that authors integrate the new data obtained on CD146-ECKO mice in the

supplementary data (figures R6 and R7) and have a short discussion about them in the paper.

Response: We are very happy that the reviewer is satisfied with the revised manuscript. We sincerely thank the reviewer for this helpful comment. As suggested, we have integrated the new data (Figures R6 and R7) into Supplementary Information and added a short discussion in the final version (page 11).